# Fixation patterns in simple choice reflect optimal information sampling

**Frederick Callaway**[1]*, **Antonio Rangel**[2], **Thomas L. Griffiths**[1,3]

**1** Department of Psychology, Princeton University, Princeton, New Jersey, United States of America,
**2** Departments of Humanities and Social Sciences and Computation and Neural Systems, California Institute of Technology, Pasadena, California, United States of America, **3** Department of Computer Science, Princeton University, Princeton, New Jersey, United States of America

* fredcallaway@princeton.edu

**Data Availability Statement:** The data and analysis code can be found on Github: https://github.com/fredcallaway/optimal-fixations-simple-choice.

**Funding:** This research was supported by a grant from Facebook Reality Labs awarded to TG

## Abstract

Simple choices (e.g., eating an apple vs. an orange) are made by integrating noisy evidence that is sampled over time and influenced by visual attention; as a result, fluctuations in visual attention can affect choices. But what determines what is fixated and when? To address this question, we model the decision process for simple choice as an information sampling problem, and approximate the optimal sampling policy. We find that it is optimal to sample from options whose value estimates are both high and uncertain. Furthermore, the optimal policy provides a reasonable account of fixations and choices in binary and trinary simple choice, as well as the differences between the two cases. Overall, the results show that the fixation process during simple choice is influenced dynamically by the value estimates computed during the decision process, in a manner consistent with optimal information sampling.

## Author summary

Any supermarket shopper is familiar with the problem of choosing between a small number of items. Even these "simple choices" can be challenging because we have to think about the options to determine which one we like most, and we can't think about all of them at once. This raises a question: what should we think about—and for how long should we think—before making a decision? We formalize this question as an information sampling problem, and identify an optimal solution. Observing what people look at while making choices, we find that many of the key patterns in their eye fixations are consistent with optimal information sampling.

## Introduction

Consider the problems faced by a diner at a buffet table or a shopper at a supermarket shelf. They are presented with a number of options and must evaluate them until they identify the most desirable one. A central question in psychology and neuroscience is to understand the algorithms, or computational processes, behind these canonical simple choices.

(https://research.fb.com/category/augmented-reality-virtual-reality/) and a grant from the NOMIS Foundation (https://nomisfoundation.ch/) awarded to AR. The funders had no role in study design, data collection and analysis, decision to publish, or preparation of the manuscript.

**Competing interests:** The authors have declared that no competing interests exist.

Previous work has established two important features of the processes underlying simple value-based choices. First, choices and reaction times are well explained by information sampling models like the diffusion decision model (DDM) [1–3] and the leaky competing accumulator model [4, 5]. In these models, individuals are initially uncertain about the desirability of each option, but they receive noisy signals about the options' values that they integrate over time to form more accurate estimates. A central insight of these models is that sampling information about unknown subjective values is a central feature of simple choice. Second, visual attention affects the decision-making process. In particular, items that are fixated longer are more likely to be chosen [6–13], unless they are aversive, in which case they are chosen *less* frequently [7, 14]. These findings have been explained by the Attentional Drift Diffusion Model (aDDM), in which the value samples of the fixated item are over-weighted relative to those of unfixated ones (or equivalently in the binary case, discounting the influence of the unattended item on the drift rate) [9, 10, 12, 13]. See [15, 16] for reviews.

These insights raise an important question: What determines what is fixated and when during the decision process? Previous work has focused on two broad classes of theories. One class suggests that decisions and fixations are driven by separate processes, so that fixations affect how information about values is sampled and integrated, but not the other way around. In this view, although fixations can be modulated by features like visual saliency or spatial location, they are assumed to be independent of the state of the decision process. This is the framework behind the aDDM [9, 10, 12] and related models [17–19].

Another class of theories explores the idea that the decision process affects fixations, especially after some information about the options' values has been accumulated. Examples of this class include the Gaze Cascade Model [6], an extension of the aDDM in which options with more accumulated evidence in their favor are more likely to be fixated [20], and a Bayesian sampling model in which options with less certain estimates are more likely to be fixated [21]. However, these models have not considered how uncertainty and value might interact, nor have they considered the optimality of the posited fixation process (although see [22–24] for such analyses in simplified settings).

Research on eye movements in the perceptual domain suggests a third possibility: that fixations are deployed to sample information optimally in order to make the best choice. Previous work in vision has shown that fixations are guided to locations that provide useful information for performing a task, and often in ways that are consistent with optimal sampling [25]. For example, in visual search (e.g., finding an 'M' in a field of 'Ns') people fixate on areas most likely to contain the target [26, 27]; in perceptual discrimination problems, people adapt their relative fixation time to the targets' noise levels [28, 29]; and in naturalistic task-free viewing, fixations are drawn to areas that have high "Bayesian surprise", i.e., areas where meaningful information is most likely to be found [30]. The properties of fixations in these types of tasks are captured by optimal sampling models that maximize expected information gain [25, 31]. However, these models have not been applied in the context of value-based decision making, and thus the extent to which fixation patterns during simple choices are consistent with optimal information sampling is an open question.

In this paper, we draw these threads together by defining a model of optimal information sampling in canonical simple choice tasks and investigating the extent to which it accounts for fixation patterns and their relation to choices. In a value-based choice, optimal information sampling requires maximizing the difference between the value of the chosen item and the cost of acquiring the information needed to make the choice. Our model thus falls into a broad class of models that extend classical rational models of economic choice [32, 33] to additionally account for constraints imposed by limited cognitive resources [34–39]. However, as is common in this approach, we stop short of specifying a full algorithmic model of simple choice.

Instead, we ask to what extent people's fixations are consistent with optimal information sampling, without specifying how the brain actually implements an optimal sampling policy.

Exploring an optimal information sampling model of fixations in simple choice is useful for several reasons. First, since fixations can affect choices, understanding what drives the fixation process can provide critical insight into the sources of mistakes and biases in decision-making. In particular, the extent to which behaviors can be characterized as mistakes depends on the extent to which fixations sample information sub-optimally. Second, simple choice algorithms like the DDM have been shown to implement optimal Bayesian information processing when the decision-maker receives the same amount of information about all options at the same rate [40–46], and this is often viewed as an explanation for why the brain uses these algorithms in the first place. In contrast, the optimal algorithm when the decision-maker must sample information selectively is unknown. Third, given the body of evidence showing that fixations are deployed optimally in perceptual decision making, it is interesting to ask if the same holds for value-based decisions. Given that such problems are characterized by both a different objective function (maximizing a scalar value rather than accuracy) and a different source of information (e.g., sampling from memory [47–49] rather than from a noisy visual stimulus), it is far from clear that optimal information sampling models will still provide a good account of fixations in this setting.

Building on the previous literature, our model assumes that the decision maker estimates the value of each item in the choice set based on a sequence of noisy samples of the items' true values. We additionally assume that these samples can only be obtained from the attended item, and that it is costly to take samples and to switch fixation locations. This sets up a sequential decision problem: at each moment the decision maker must decide whether to keep sampling, and if so, which item to sample from. Since the model does not have a tractable analytical solution, in order to solve it and take it to the data, we approximate the optimal solution using tools from metareasoning in artificial intelligence [50–53].

We compare the optimal fixation policy to human fixation patterns in two influential binary and trinary choice datasets [9, 10]. We find that the model captures many previously identified patterns in the fixation data, including the effects of previous fixation time [21] and item value [17, 20, 22]. In addition, the model makes several novel predictions about the differences in fixations between binary and trinary choices and about fixation durations, which are consistent with the data. Finally, we identify a critical role of the prior distribution in producing the classic effects of attention on choice [7, 9, 10, 14]. Overall, the results show that the fixation process during simple choice is influenced by the value estimates computed during the decision process, in a manner consistent with optimal information sampling.

## Model

### Sequential sampling model

We consider simple choice problems in which a decision maker (DM) is presented with a set of items (e.g., snacks) and must choose one. Each item $i$ is associated with some true but unknown value, $u^{(i)}$, the utility that the DM would gain by choosing it. Following previous work [1, 9, 40, 42–46, 54], we assume that the DM informs her choice by collecting noisy samples of the items' true values, each providing a small amount of information, but incurring a small cost. The DM integrates the samples into posterior beliefs about each item's value, choosing the item with maximal posterior mean when she terminates the sampling process.

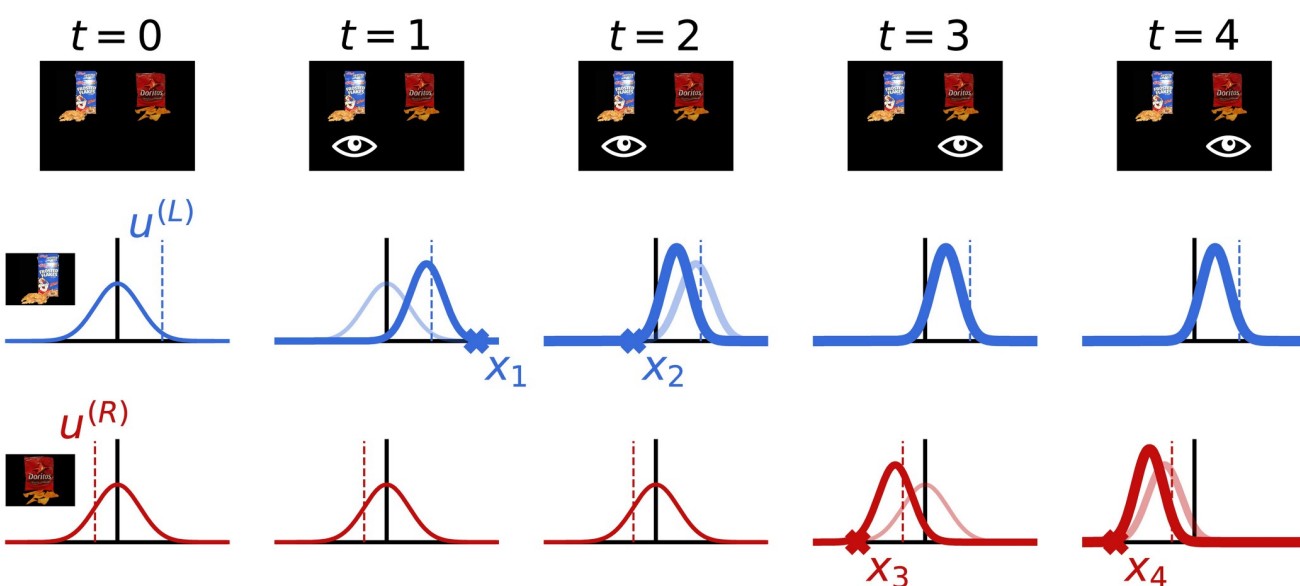

**Fig 1. Sampling and belief updating in the binary choice task.** The top row shows the experimental display, with the fixated item denoted by the eye symbol. The bottom two rows depict the first few steps of the sampling and belief updating process. The decision maker's beliefs about the value of each item are denoted by the Gaussian probability density curves. The true values of each item (dashed lines) are sampled from standard normal distributions; this is captured in the decision maker's initial belief state (first column). Every time step, $t$, the decision maker fixates one of the items and receives a noisy sample about the true value of that item ($x_t$ marks). She then updates her belief about the value of the fixated item using Bayesian updating (shift from light to dark curve). The beliefs for the unfixated item are not updated. The process repeats each time step until the decision maker terminates sampling, at which point she chooses the item with maximal posterior mean.

As illustrated in Fig 1, we model attention by assuming that the DM can only sample from one item at each time point, the item she is fixating on. This sets up a fundamental problem: How should she allocate fixations in order to make good decisions without incurring too much cost? Specifically, at each time point, the DM must decide whether to select an option or continue sampling, and in the latter case, she must also decide which item to sample from. Importantly, she cannot simply allocate her attention to the item with the highest true value because she does not know the true values. Rather, she must decide which item to attend to based on her current value estimates and their uncertainty.

The DM's belief about the item values at time $t$ is described by a set of Gaussians, one for each item, with means $\mu_t^{(i)}$ and precisions $\lambda_t^{(i)}$ (the precision is the inverse of the variance). These estimated value distributions are initialized to the DM's prior belief about the distribution of values in the environment. That is, she assumes that $u^{(i)} \sim \text{Gaussian}(\bar{\mu}, \bar{\sigma}^2)$ and consequently sets $\mu_0^{(i)} = \bar{\mu}$ and $\lambda_0^{(i)} = \bar{\sigma}^{-2}$ for all $i$. We further discuss the important role of the prior below.

We model the control of attention as the selection of cognitive operations, $c_t$, that specify either an item to sample, or the termination of sampling. If the DM wishes to sample from item $c$ at time-step $t$, she selects $c_t = c$ and receives a signal

$$x_t \sim \text{Gaussian}(u^{(c)}, \sigma_x^2), \qquad (1)$$

where $u^{(c)}$ is the *unknown* true value of the item being sampled, and $\sigma_x^2$ is a free parameter specifying the amount of noise in each signal. The belief state is then updated in accordance with

Bayesian inference:

$$\begin{aligned}
\lambda_{t+1}^{(c)} &= \lambda_t^{(c)} + \sigma_x^{-2} \\
\mu_{t+1}^{(c)} &= \frac{\sigma_x^{-2} x_t + \lambda_t^{(c)} \mu_t^{(c)}}{\lambda_{t+1}^{(c)}} \\
\lambda_{t+1}^{(i)} &= \lambda_t^{(i)} \text{ and } \mu_{t+1}^{(i)} = \mu_t^{(i)} \text{ for } i \neq c.
\end{aligned} \tag{2}$$

The cognitive cost of each step of sampling and updating is given by a free parameter, $\gamma_{\text{sample}}$. We additionally impose a switching cost, $\gamma_{\text{switch}}$, that the DM incurs whenever she samples from an item other than the one sampled on the last timestep (i.e., makes a saccade to a different item). Thus, the cost of sampling is

$$\text{cost}(c_t) = \gamma_{\text{sample}} + \mathbf{1}(c_t \neq c_{t-1})\, \gamma_{\text{switch}}. \tag{3}$$

Note that the model includes the special case in which there are no switching costs ($\gamma_{\text{switch}} = 0$).

In addition to choosing an item to sample, the DM can also decide to stop sampling and choose the item with the highest expected value. In this case, she selects $c_t = \bot$. It follows that if the choice is made at time step $T$ (i.e., $c_T = \bot$) the chosen item is $i^* = \arg\max_i \mu_T^{(i)}$. The DM's total payoff on a single decision is given by:

$$\text{payoff} = \underbrace{u^{(i^*)}}_{\substack{\text{utility of} \\ \text{chosen item}}} - \underbrace{\sum_{t=1}^{T-1} \text{cost}(c_t)}_{\text{cognitive cost}}. \tag{4}$$

## Optimal policy

We assume that the decisions about where to fixate and when to stop sampling are made optimally, subject to the informational constraints described in the previous section. Formally, we assume that the $c_t$ are selected by an *optimal policy*. A policy selects the next cognitive operation to execute, $c_t$, given the current belief state, $(\mu_t, \lambda_t)$; it is optimal if it selects $c_t$ in a way that maximizes the expectation of Eq 4. How can we identify such a policy? Problems of this kind have been explored in the artificial intelligence literature on rational metareasoning [50, 51]. Thus, we cast the model described above as a metalevel Markov decision process [52], and identify a near-optimal policy using a recently developed method that has been shown to achieve strong performance on a related problem [53]. In accordance with past work modeling people's choices [55] and fixations [20, 21], we assume that people follow a softmax policy in selecting each cognitive operation by sampling from a Boltzmann distribution based on their estimated values. Thus, their choices of cognitive operations are guided by the optimal policy, but subject to some noise. See Methods for details.

What does optimal attention allocation look like? In order to provide an intuitive understanding, we focus on two key properties of belief states: (1) uncertainty about the true values and (2) differences in the value estimates. Fig 2A shows the probability of the optimal policy (for a model with parameters fit to human data) sampling an item as a function of these two dimensions (marginalizing over the other dimensions according to their probability of occurring in simulated trials). We see that the optimal policy tends to fixate on items that are

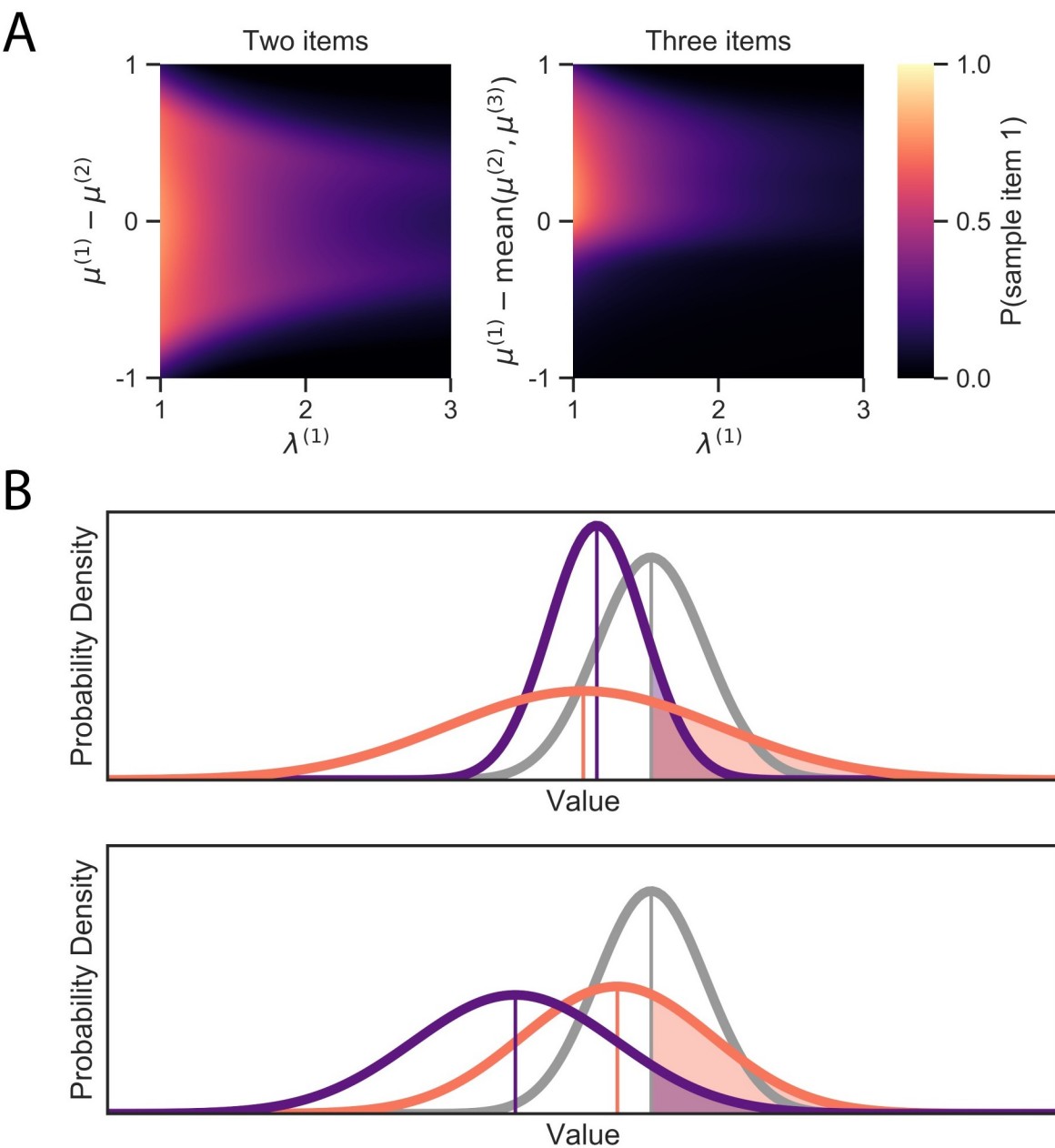

**Fig 2. Optimal fixation policy.** (A) Probability of fixating on item 1 as a function of the precision of its value estimate, $\lambda^{(1)}$, and the mean of its relative value estimate, $\mu^{(1)} - \text{mean}(\mu^{(2)}, \mu^{(3)})$. The heat map denotes the probability of fixating item 1 as opposed to fixating one of the other items or terminating the sampling process. (B) Illustration of the value of sampling. Each panel shows a belief state for trinary choice. The curves depict the estimated beliefs for each item's value, and the shaded regions show the probability that the item's true value is higher than the current best value estimate. This probability correlates strongly with the value of sampling the item because sampling is only valuable if it changes the choice (the full value of sampling additionally depends on the size of the potential gain in value, as well as the cost of future samples and the possibility of sampling other items). In each case, it is more valuable to sample the orange item than the purple item because either (top) its value is more uncertain, or (bottom) its value is closer to the leading value.

uncertain and have estimated values similar to the other items. In the case of trinary—but not binary—choice, we additionally see a stark asymmetry in the effect of relative estimated value. While the policy is likely to sample from an item whose value is substantially higher than the competitors, it is unlikely to sample from an item with value well below. In

particular, the policy has a strong preference to sample from the items with best or second-best value estimates.

To see why this is optimal, note that sampling is only valuable insofar as it affects choice, and that the chosen item is the one with maximal estimated value when sampling stops. Thus, the optimal policy generally fixates on the item for which gathering more evidence is most likely to change which item has maximal expected value. There are two ways for this to happen: either the value of the current best item is reduced below the second-best item, or the value of some alternative item is increased above the best item. The former can only happen by sampling the best item, and the latter is *ceteris paribus* most likely to occur by sampling the second-best item because it is closer to the top position than the third-best item is (Fig 2B bottom). However, if uncertainty is much greater for the third-best item, this can outweigh the larger difference in estimated value (Fig 2B top). See [22] for a more formal justification for value-directed attention in a simplified non-dynamic case.

## The prior distribution

Recall that the initial belief about each item's value is set to the DM's prior belief about the distribution of values in the environment; that is $\mu_0^{(i)} = \bar{\mu}$ and $\lambda_0^{(i)} = \bar{\sigma}^{-2}$. This corresponds to the DM assuming that each item's value is drawn from a prior distribution of true values given by $u^{(i)} \sim \mathrm{Gaussian}(\bar{\mu}, \bar{\sigma}^2)$. This assumption is plausible if this is the actual distribution of items that the DM encounters, and she is a Bayesian learner with sufficient experience in the context under study. However, given that these models are typically used to study choices made in the context of an experiment (as we do here), the DM might not have learned the exact prior distribution at work. As a result, we must consider the possibility that she has a *biased prior*.

In order to investigate the role of the prior on the model predictions, we assume that it takes the form of a Gaussian distribution with a mean and standard deviation related to the actual empirical distribution as follows:

$$\begin{aligned}
\bar{\mu} &= \alpha \cdot \mathrm{mean}\,(\mathrm{ratings}) \\
\bar{\sigma} &= \mathrm{std}\,(\mathrm{ratings}).
\end{aligned} \tag{5}$$

Here, mean(ratings) denotes the mean value ratings of all items, which provide independent and unbiased measures of the true value of the items (computed across trials in both experiments), and $\alpha$ is a free parameter that specifies the amount of bias in the prior ($\alpha = 0$ corresponds to a strong bias and $\alpha = 1$ corresponds to no bias). As a result, the DM has correct beliefs about the prior variance, but is allowed to have a biased belief about the prior mean. This case could arise, for example, if the average true value of the items used in the experiment differs from the average item that the DM encounters in her daily life.

## Model fitting

We apply the model to two influential simple choice datasets: a binary food choice task [9] and a trinary food choice task [10]. In each study, participants first provided liking-ratings for 70 snack items on a -10 to 10 scale, which are used as an independent measure of the items' true values. They then made 100 choices among items that they had rated positively, while the location of their fixations was monitored at a rate of 50 Hz. See S1 Appendix for more details on the experiments.

The model has five free parameters: the standard deviation of the sampling distribution $\sigma_x$, the cost per sample $\gamma_{\mathrm{sample}}$, the cost of switching attention $\gamma_{\mathrm{switch}}$, the prior bias $\alpha$, and the inverse temperature of the softmax policy used to select cognitive operations, $\beta$. This last

parameter controls the amount of noise in the fixation decisions. In order to fit the model, we need to make an assumption about the time that it takes to acquire each sample, which we take to be 100 ms. Note, however, that this choice is not important: changing the assumed duration leads to a change in the fitted parameters, but not in the qualitative model predictions.

We use an approximate maximum likelihood method to fit these parameters to choice and fixation data, which is described in the Methods section. Importantly, since the same model can be applied to N-item choices, we fit a common set of parameters jointly to the pooled data in both datasets. Thus, any differences in model predictions between binary and trinary choices are *a priori* predictions resulting from the structure of the model, and not differences in the parameters used to explain the two types of choices. We estimate the parameters using only the even trials, and then simulate the model in odd trials in order to compare the model predictions with the observed patterns out-of-sample. Because the policy optimization and likelihood estimation methods that we use are stochastic, we display simulations using the 30 top performing parameter configurations to give a sense of the uncertainty in the predictions. The parameter estimates were (mean ± std) $\sigma_x = 2.60 \pm 0.216$, $\alpha = 0.581 \pm 0.118$, $\gamma_{\text{switch}} = 0.00995 \pm 0.001$, $\gamma_{\text{sample}} = 0.00373 \pm 0.001$, and $\beta = 364 \pm 81.2$ As explained in the Methods, the units of these parameter estimates are standard deviations of value (i.e., $\bar{\sigma}$).

In order to explore the role of the prior, we also fit versions of the model in which the prior bias term was fixed to $\alpha = 0$ or $\alpha = 1$. The former corresponds to a strongly biased prior and the latter corresponds to a completely unbiased prior. For $\alpha = 0$, the fitted parameters were $\sigma_x = 3.16 \pm 0.409$, $\gamma_{\text{switch}} = 0.00875 \pm 0.002$, $\gamma_{\text{sample}} = 0.00319 \pm 0.001$, and $\beta = 326 \pm 81.2$. For $\alpha = 1$, they were $\sigma_x = 2.66 \pm 0.272$, $\gamma_{\text{switch}} = 0.0118 \pm 0.002$, $\gamma_{\text{sample}} = 0.00506 \pm 0.001$, and $\beta = 330.0 \pm 97.9$.

All the figures below are based on model fits estimated at the group level on the pooled data. However, for completeness we also fit the model separately for each individual, and report these fits in S2 Appendix. We also carry out a validation of our model fitting approach in S1 Appendix.

## Results

We now investigate the extent to which the predictions of the model, fitted on the even trials, are able to account for observed choice, reaction time and fixation patterns in the out-of-sample odd trials.

### Basic psychometrics

We begin by looking at basic psychometric patterns. Fig 3A compares the choice curves predicted by the model with the actual observed choices, separately for the case of binary and trinary choice. It shows that the model captures well the influence of the items' true values (as measured by liking ratings) on choice.

Fig 3B plots the distribution of total fixation times. This measure is similar to reaction time except that it excludes time not spent fixating on one of the items. We use total fixation time instead of reaction time because the model does not account for the initial fixation latency nor the time spent saccading between items (although it does account for the opportunity cost of that time, through the $\gamma_{\text{sample}}$ parameter). As shown in the figure, the model provides a reasonable qualitative account of the distributions, although it underpredicts the mode in the case of two items and the skew in both cases.

Fig 3C shows the relationship between total fixation time and trial difficulty, as measured by the relative liking rating of the best item. We find that the model provides a reasonable account of how total fixation time changes with difficulty. This prediction follows from the

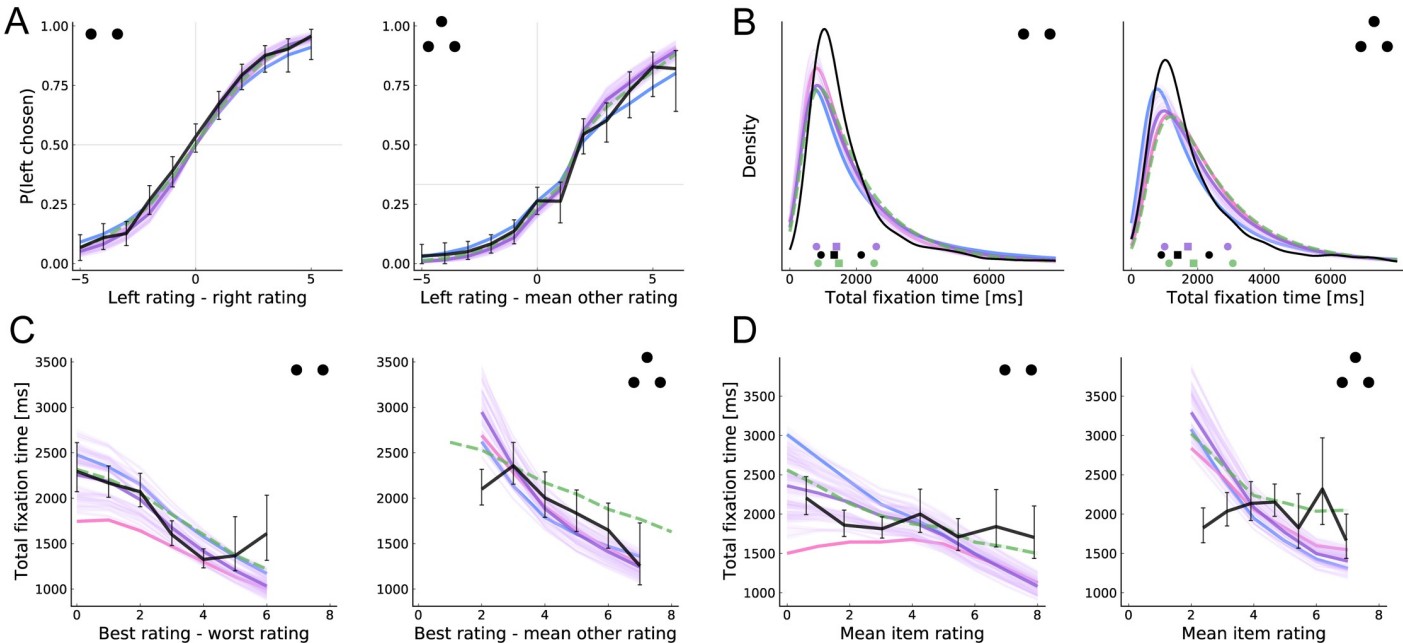

**Fig 3. Basic psychometrics.** Each panel compares human data (black) and model predictions for binary choice (left, two dots) and trinary choice (right, three dots). The main model predictions are shown in purple. The restricted model predictions for the case of a highly biased prior mean ($\alpha = 0$) are shown in blue; the case of a highly unbiased prior mean ($\alpha = 1$) is shown in pink. These colors were chosen to illustrate that the main model falls between these two extremes. The aDDM predictions are shown in dashed green. Error bars (human) and shaded regions (model) indicate 95% confidence intervals computed by 10,000 bootstrap samples (the model confidence intervals are often too small to be visible). Note that the method used to compute and estimate the model parameters is noisy. To provide a sense of the effect of this noise on the main model predictions, we depict the predictions of the thirty best-fitting parameter configurations. Each light purple line depicts the predictions for one of those parameters, whereas the darker purple line shows the mean prediction. In order to keep the plot legible, only the mean predictions of the biased priors models are shown. (A) Choice probability as a function of relative rating. (B) Kernel density estimation for the distribution of total fixation time. Quartiles (25%, 50%, and 75% quantiles) for the data, aDDM and main model predictions are shown at the bottom. (C) Total fixation time as a function of the relative rating of the highest rated item. (D) Total fixation time as a function of the mean of all the item ratings (overall value).

fact that fewer samples are necessary to detect a large difference than to either detect a small difference or determine that the difference is small enough to be unimportant. However, the model exhibits considerable variation in the predicted intercept and substantially overpredicts total fixation time in difficult trinary choices.

Finally, Fig 3D shows the relationship between total fixation time and the average rating of all the items in the choice set. This "overall value effect" has been emphasized in recent research [13, 16] because it is consistent with multiplicative attention weighting (as in the aDDM) but not an additive boosting model (e.g., [11]). Bayesian updating results in a form of multiplicative weighting (specifically, a hyperbolic function, c.f. [14]), and thus our model also predicts this pattern. Surprisingly, we do not see strong evidence for the overall value effect in the datasets we consider, but we note that the effect has been found robustly in several other datasets [13, 56–59]. Note that, in the binary case, the predicted overall value effect is symmetric around the prior mean; that is, choices between two very bad items will also be made quickly. Indeed, with an unbiased-prior, the model predicts an inverted-U relationship around the prior mean.

Several additional patterns in Fig 3 are worth highlighting. First, all the models make similar and reasonable predictions of the psychometric choice curve and fixation time distributions. Second, the models with some prior bias provide a better account of the fixation time curves in binary choice than the unbiased model, and qualitatively similar predictions to the aDDM.

Finally, despite using a common set of parameters, all the models capture well the differences between binary and trinary choice.

## Basic fixation properties

We next compare the predicted and observed fixation patterns. An observed "fixation" refers to a contiguous span of time during which a participant looks at the same item. A predicted model fixation refers to a continuous sequence of samples taken from one item.

Fig 4A shows the distribution of the number of fixations across trials. The model-predicted distribution is reasonably similar to the observed data. However, in the two-item case, the model is more likely to make only one fixation, suggesting that people have a tendency to fixate both items at least once that the model does not capture.

Fig 4B shows the relationship between the total number of fixations and decision difficulty. We find that the model captures the relationship between difficulty and the number of fixations reasonably well, with the same caveats as for Fig 3B.

The original binary and trinary choice papers [9, 10] observed a systematic change in fixation durations over the course of the trial, as shown in Fig 4C. Although the model tends to underpredict the duration of the first two fixations in the three-item case, it captures well three key patterns: (a) the final fixation is shorter, (b) later (but non-final) fixations are longer and (c) fixations are substantially longer in the two-item case. The final prediction is especially striking given that the model uses the same set of fitted parameters for both datasets. The model predicts shorter final fixations because they are cut off when a choice is made [9, 10]. The model predicts the other patterns because more evidence is needed to alter beliefs when their precision is already high; this occurs late in the trial, especially in the two-item case where samples are split between fewer items.

Fig 4 also shows that the main model provides a more accurate account than the aDDM of how the number of fixations changes with trial difficulty, and of how fixation duration evolves

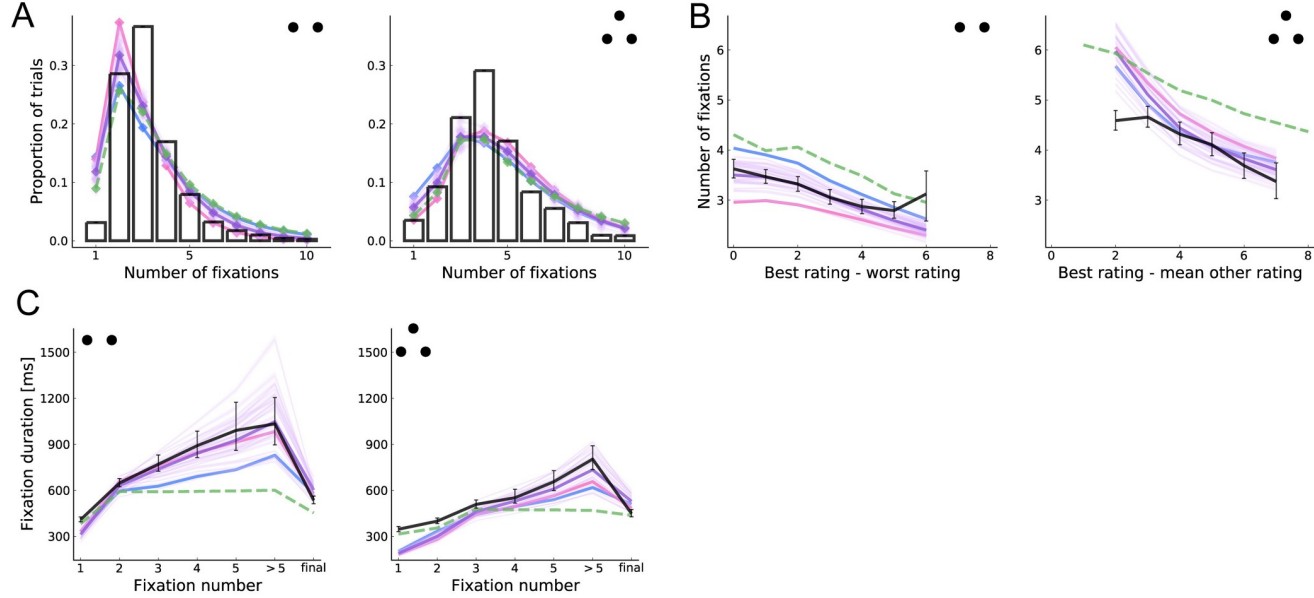

**Fig 4. Basic fixation patterns.** (A) Histogram of number of fixations in a trial. (B) Number of fixations as a function of decision difficulty, as measured by the relative rating of the best item. (C) Duration of fixation by fixation number. Final fixations are excluded from all but the last bin. See Fig 3 for more details.

over the course of a trial. One difficulty in making this comparison is that the aDDM assumes that non-final fixation durations are sampled from the observed empirical distribution, conditional on a number of observable variables, and thus the accuracy of its predictions regarding fixation duration and fixation number depends on the details of this sampling. To maximize comparability with the existing literature, here we use the same methods as in the original implementations [9, 10].

## Uncertainty-directed attention

As we have seen, one of the key drivers of fixations in the optimal policy is uncertainty about the items' values. Specifically, because the precision of the posteriors increases linearly with the number of samples, the model predicts that, other things being equal, fixations should go to items that have received less cumulative fixation time. However, the difference in precision must be large enough to justify paying the switching cost. In this section we explore some of the fixation patterns associated with this mechanism.

Fig 5A depicts the distribution of relative cumulative fixation time at the beginning of a new fixation, starting with the second fixation. That is, at the onset of each fixation, we ask how much time has already been spent fixating the newly fixated item, compared to the other items. In both cases, the actual and predicted distributions are centered below zero, so that items tend to be fixated when they have received less fixation time than the other items. Additionally, the model correctly predicts the lower mode and fatter left tail in the two-item case.

Note, however, that a purely mechanical effect can account for this basic pattern: the item that is currently fixated will on average have received the most fixation time, but it cannot be the target of a new fixation, which drives down the fixation advantage of newly fixated items. For this reason, it is useful to look further at the three-item case, which affords a stronger test of uncertainty-directed attention. In this case, the target of each new fixation (excluding the first) must be one of the two items that are not currently fixated. Thus, comparing the cumulative fixation times for these items avoids the previous confound. Fig 5B thus plots the distribution of fixation time for the fixated item minus that of the item which could have been fixated but was not. We see a similar pattern to Fig 5A (right) in both the data and model predictions. This suggests that uncertainty is not simply driving the decision to make a saccade, but is also influencing the location of that saccade.

Fig 5C explores this further by looking at the location of new fixations in the three-item case, as a function of the difference in cumulative fixation time between the two possible fixation targets. Although the more-previously-fixated item is always less likely to be fixated, the

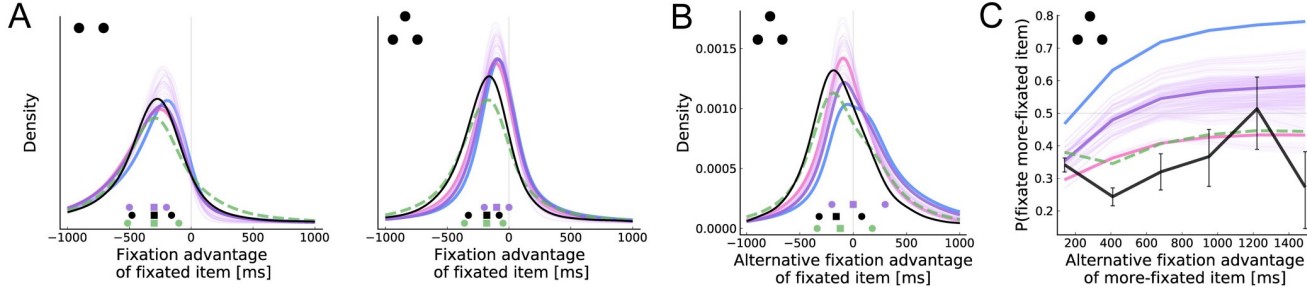

**Fig 5. Uncertainty-directed attention.** (A) Distribution of fixation advantage of the fixated item, computed at the beginning of each new fixation. Fixation advantage is defined as the cumulative fixation time to the item minus the mean cumulative fixation time to the other item(s). First fixations are excluded in this plot. (B) Similar to A, except that we compare the fixation advantage between the fixated item and the other item that could have been fixated but was not. First and second fixations are excluded in this plot. (C) The probability that the item with greater alternative fixation advantage is fixated, as a function of that advantage. See Fig 3 for more details.

probability of such a fixation actually *increases* as its fixation advantage grows. This counterintuitive model prediction results from the competing effects of value and uncertainty on attention. Since items with high estimated value are fixated more, an item that has been fixated much less than the others is likely to have a lower estimated value, and is therefore less likely to receive more fixations. However, we see that the predicted effect is much stronger than the observed effect, and that the aDDM model provides a better account of this pattern than our main model. However, note that the accuracy of this fit follows from the fact that the aDDM samples fixation locations and durations from the empirical distribution, conditioned on the previous three fixation locations and the item ratings.

## Value-directed attention

A second key driver of attention in the optimal policy is estimated value, which directs fixations to the *two* items with the highest posterior means. As illustrated in Fig 2A, this implies that fixation locations should be sensitive to relative estimated values in the trinary but not in the binary case.

Although we cannot directly measure the participants' evolving value estimates, we can use the liking ratings as a proxy for them because higher-rated items will tend to result in higher value estimates. Using this idea, Fig 6A shows the proportion of fixation time devoted to the left item as a function of its relative rating. Focusing first on the three-item case, both the model and data show a strong tendency to spend more time fixating on higher rated items (which are therefore likely to have higher estimated values). In the two item case, the model simulations show a smaller but also positive effect. This is counterintuitive since the model predicts that in the two-item case fixation locations are insensitive to the sign of the relative value estimates (Fig 2A). However, the pattern likely arises due to the tendency to fixate last on the chosen item (see Fig 7A below).

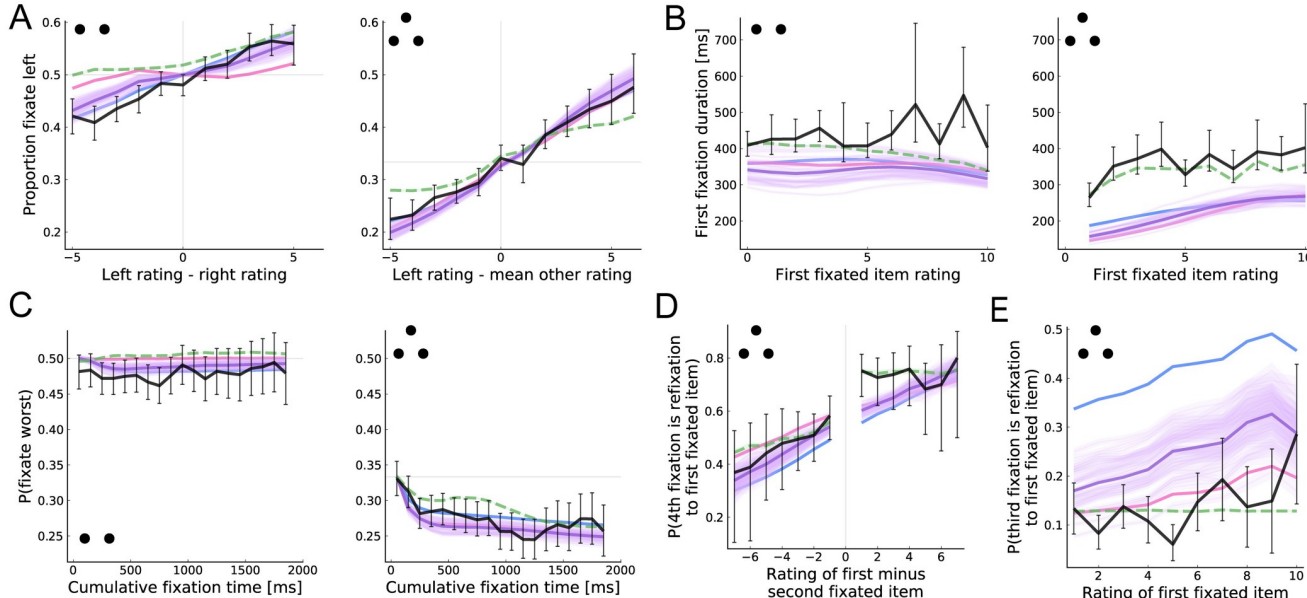

**Fig 6. Value-directed attention.** (A) Proportion of time fixating the left item as a function of its relative rating. (B) First fixation duration as a function of the rating of the first-fixated item. (C) Probability of fixating the lowest rated item as a function of the cumulative fixation time to any of the items. (D) Probability that the fourth fixation is to the first-fixated item as a function of the difference in rating between that item and the second-fixated item. (E) Probability that the third fixation is to the first fixated item as a function of its rating. See Fig 3 for more details.

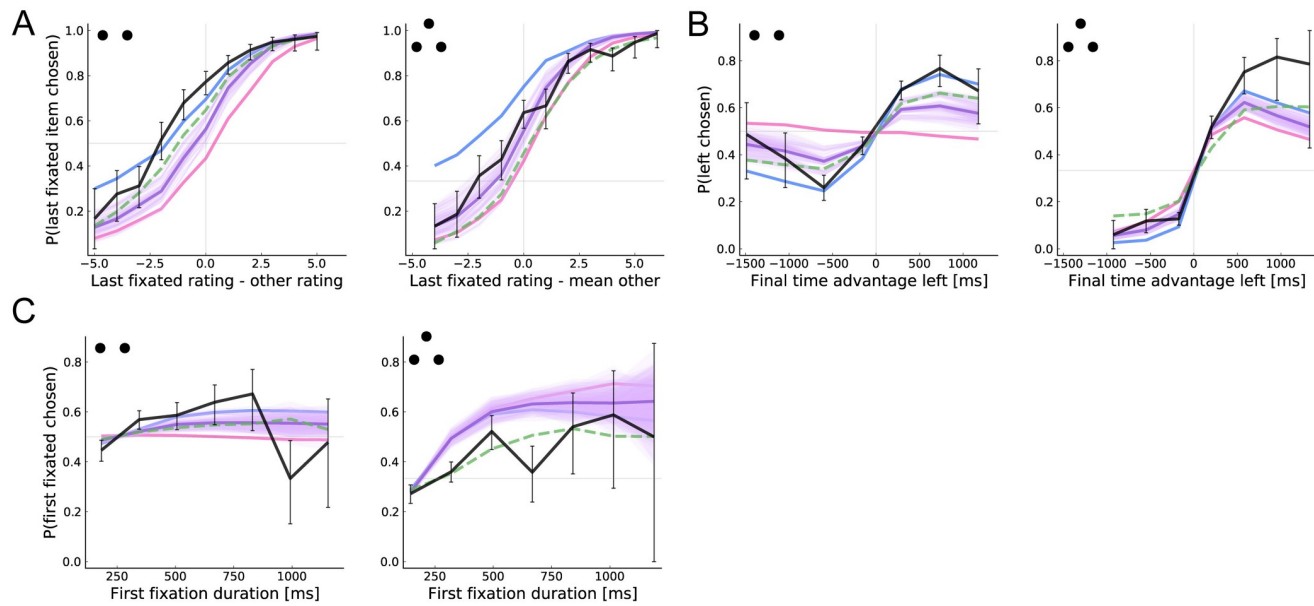

**Fig 7. Choice biases.** (A) Probability that the last fixated item is chosen as a function of its relative rating. (B) Probability that the left item is chosen as a function of its final fixation advantage, given by total fixation time to the left item minus the mean total fixation time to the other item(s). (C) Probability of choosing the first-seen item as a function of the first-fixation duration. See Fig 3 for more details.

Fig 6B provides an alternative test that avoids confounds associated with the final fixation. It shows the duration of the first fixation, which is rarely final, as a function of the rating of the first fixated item. In the three-item case, both the model and data show longer initial fixations to high-rated items, although the model systematically underpredicts the mean first fixation duration. This prediction follows from the fact that, under the optimal policy, fixations are terminated when the fixated item's estimated value falls out of the top two (below zero for the first fixation); the higher the true value of the item, the less likely this is to happen. In the two-item case, however, the model predicts that first fixation duration should be largely insensitive to estimated value; highly valuable items actually receive slightly *shorter* fixations because these items are more likely to generate extremely positive samples that result in terminating the first fixation and immediately choosing the fixated item. Consistent with this prediction, humans show little evidence for longer first fixations to high-rated items in the binary case.

Previous work has suggested that attention may be directly influenced by the true value of the items [17, 18, 60]. In our model, however, attention is driven only by the internal value estimates generated during the decision making process. To distinguish between these two accounts, we need a way to dissociate estimated value from true value. One way to do this is by looking at the time course of attention. Early in the decision making process, estimated values will be only weakly related to true value. However, with time the value estimates become increasingly accurate and thus more closely correlate with true value. Thus, if the decision maker always attends to the items with high *estimated* value, she should be increasingly likely to attend to items with high *true* value as the trial progresses. Fig 6C shows the probability of fixating on the worst item as a function of the cumulative fixation time to any of the items. In both the two- and three-item cases, the probability begins near chance. In the three-item case, however, the probability quickly falls. This is consistent with a model in which attention is driven by estimated value rather than value itself.

The model makes even starker predictions in the three-item case. First, take all trials in which the decision-maker samples from different items during the first three fixations. Consider the choice of where to deploy the fourth fixation. The model predicts that this fixation should be to the first-fixated item if its posterior mean is larger than that of the second-fixated item, and vice versa. As a result, the probability that the fourth fixation is a refixation to the first-fixated item should increase with the difference in ratings between the first- and second-fixated items. As shown in Fig 6D, the observed pattern follows the model prediction.

Finally, the model makes a striking prediction regarding the location of the third fixation in the three-item case. Consider the choice of where to fixate after the first two fixations. The decision maker can choose to fixate on the item that she has not seen yet, or to refixate the first-fixated item. The model predicts a refixation to the first-seen item if both that item and the second-seen item already have high value estimates (leaving the unfixated item with the lowest value estimate). Consistent with this prediction, Fig 6E shows that the probability of the third fixation being a refixation to the first-seen item increases with that item's rating. Note that the model with $\alpha$ fixed to zero (corresponding to a strong prior bias), dramatically over-predicts the intercept. This is because this model greatly underestimates the value of the not-yet-fixated item.

Fig 6 shows that our main model provides a better prediction of some fixation patterns, whereas the aDDM provides a better fit of others. However, it is important to keep in mind that whereas our model provides predictions for these fixation patterns based on first principles, the predictions of the aDDM for these patterns are largely mechanistic since that model samples fixation locations and durations from the observed empirical distribution. As a result, it is not surprising that Fig 6B shows a better match between the aDDM and the data since the predicted durations are, literally, sampled from the observed data conditional on the first item rating.

## Choice biases

Previous work has found a systematic positive correlation between relative fixation time and choice for appetitive (i.e., positively valenced) items [6, 7, 9, 10, 14, 20]. In particular, models like the aDDM propose that an exogenous or random increase in fixations towards an appetitive item increase the probability that it will be chosen, which leads to attention driven choice biases. Here we investigate whether the optimal model can account for these types of effects.

Importantly, in the type of optimal fixation model proposed here, there are two potential mechanisms through which such correlations can emerge. The first is driven by the prior. If the prior mean is negatively biased, then sampling from an item will on average increase its estimated value. This follows from the fact that sampling will generally move the estimated value towards the item's true value, and a negatively biased prior implies that the initial value estimate is generally less than the true value. The second mechanism, which is only present in trinary choice, is the result of value-directed attention. Here, the causal direction is flipped, with value estimates driving fixations rather than fixations driving value estimates. In particular, items with higher estimated value are both more likely to be fixated, and more likely to be chosen. Thus, fixations and choice are correlated through a common cause structure. Importantly, the two mechanisms are not mutually exclusive; in fact, our model predicts that both will be in effect for choice between more than two items.

Fig 7A shows that there is a sizable choice bias towards the last-seen item in both datasets, as evidenced by the greater-than-chance probability of choosing an item whose value is equal to the mean of the other items. Our model provides a strong quantitative account of the pattern in trinary choice, but substantially underpredicts the effect in binary choice. Interestingly,

it predicts a weaker effect than the aDDM in the binary case, but a stronger effect in the trinary case.

To understand this result, it is important to think about the prior beliefs implicit in the aDDM and related models [9, 10, 20]. Since these are not Bayesian models, they do not posit an explicit prior that is then modified by evidence. However, the aDDM can be viewed as an approximation to a Bayesian model with a prior centered on zero, as reflected by the initial point of the accumulator (zero) and the multiplicative discounting (the evidence for the non-attended item is discounted towards zero). The latter roughly corresponds to the Bayesian regularization effect, wherein the posterior mean falls closer to the prior mean when the likelihood is weak (low precision). Given this, our model predicts a weaker effect in the binary case because it has a weaker prior bias ($\alpha = 0.58$) than the one implicit in the aDDM ($\alpha = 0$). Our model predicts a stronger effect in the trinary case due to the value-directed attention mechanism. Critically, although the aDDM accounts for the effect of *true* value on fixations (by sampling from the empirical fixation distribution), only the optimal model accounts for the effects of *estimated* value. Thus, conditioning on true value (as we do in Fig 7A) breaks the value-based attention mechanism in the aDDM but not in the optimal model. Finally, note that the optimal model with $\alpha = 0$ provides a good account of the bias in the binary case, but dramatically overpredicts it in the trinary case.

Fig 7B shows that the average probability of choosing the left item increases substantially with its overall relative fixation time. As before, in comparison with the aDDM, the optimal model provides better captures the full strength of the bias in the trinary case, but underpredicts the effect in the binary case. The optimal model with $\alpha$ fixed to zero performs best in both cases. Note that the fit of the aDDM is not as close as for similar figures in the original papers because we simulate all models with the observed ratings (rather than all possible combination of item ratings) and we consider a larger range of final time advantage. We replicate the original aDDM figures in S1 Appendix.

Finally, Fig 7C shows that the probability of choosing the first fixated item increases with the duration of the first fixation. Importantly, this figure shows that the attention-choice correlation cannot be explained solely by the tendency to choose the last-fixated item. Again, all four models qualitatively capture the effect, with varying degrees of quantitative fit.

## Discussion

We have built a model of optimal information sampling during simple choice in order to investigate the extent to which it can provide a quantitative account of fixation patterns, and their relationship with choices, during binary and trinary decisions. The model is based on previous work showing that simple choices are based on the sequential accumulation of noisy value samples [1, 44, 61–64] and that the process is modulated by visual attention [7, 9, 10, 17, 20, 21, 65]. However, instead of proposing a specific algorithmic model of the fixation and choice process, as is common in the literature, our focus has been on characterizing the optimal fixation policy and its implications. We build on previous work on optimal economic decision-making in which samples are acquired for all options at the same rate [40, 44–46], and extend it to the case of endogenous attention, where the decision maker can control the rate of information acquired about each option. We formalized the selection of fixations as a problem of dynamically allocating a costly cognitive resource in order to gain information about the values of the available options. Leveraging tools from metareasoning in artificial intelligence [50–53], we approximated the optimal solution to this problem, which takes the form of a policy that selects which item to fixate at each moment and when to terminate the decision-making process.

We found that, despite its simplicity, the optimal model accounts for many key fixation and choice patterns in two influential binary and trinary choice datasets [9, 10]. The model was also able to account for striking differences between the two- and three-item cases using a common set of parameters fitted out of sample. More importantly, the results provide evidence in favor of the hypothesis that the fixation process is influenced by the evolving value estimates, at least to some extent. Consider, for example, the increase in fixation duration over the course of the trial shown in Fig 4C, the tendency to equate fixation time across items (Fig 5B), and the relationship between the rating of the first fixated item and the probability of re-fixating it (Fig 6D and 6E). These effects are explained by our model, but are hard to explain with exogenous fixations, or with fixations that are correlated with the true value of the items, but not with the evolving value estimates (e.g., as in [17, 18, 66]).

Optimal information sampling models may appear inappropriate for value-based decision-making problems, in which perceptual uncertainty about the identity of the different choice items (often highly familiar junk foods) is likely resolved long before a choice is made. Two features of the model ameliorate this concern. First, the samples underlying value-based decisions are not taken from the external display (as in perceptual decisions), but are instead generated internally, perhaps by some combination of mental simulation and memory recall [47–49]. Second, the model makes the *eye-mind* assumption [15, 67]: what a person is looking at is a good indicator of what they are thinking about. Importantly, these assumptions implicitly underlie all sequential sampling models of value-based decision-making.

Our model is not the first to propose that the fixation and value-estimation processes might interact reciprocally. However, no previous models fully capture the key characteristics of optimal attention allocation, which appear to be at least approximated in human fixation behavior. For example, the Gaze Cascade Model [6] proposes that late in a trial subjects lock-in fixations on the favored option until a choice is made, [20] propose an aDDM in which the probability of fixating an item is given by a softmax over the estimated values, and [21] propose a Bayesian model of binary choice in which fixations are driven by relative uncertainty. In contrast to these models, the optimal model predicts that fixations are driven by a combination of the estimated uncertainty and relative values throughout the trial, and that attention is devoted specifically to the items with the top two value estimates. Although the data presented here strongly support the first prediction, further data are necessary to distinguish between the top-two rule and the softmax rule of [20].

Our results shed further light on the mechanisms underlying the classic attention-choice correlation that has motivated previous models of attention-modulated simple choice. First, our results highlight an important role of prior beliefs in sequential sampling models of simple choice (c.f. [68]). All previous models have assumed a prior mean of zero, either explicitly [21, 68] or implicitly [9, 10, 20]. Such a prior is negatively biased when all or most items have positive value, as is often the case in experimental settings. This bias is critical in explaining the classic attention-choice correlation effects because it creates a net-positive effect of attention on choice: if one begins with an underestimate, attending to an item will on average increase its estimated value. However, we found that the best characterization of the full behavior was achieved with a moderately biased prior, both in terms of our approximate likelihood and in the full set of behavioral patterns in the plots.

Our results also suggest another (not mutually exclusive) mechanism by which the attention-choice correlation can emerge: value-directed attention. We found that the optimal model with no prior bias ($\alpha = 1$) predicts an attention-choice correlation in the trinary choice case. This is because, controlling for true values, an increase in estimated value (e.g., due to sample noise) makes the model more likely to both fixate and choose an item. This could potentially help to resolve the debate over additive vs. multiplicative effects of attention on

choice [11, 13]. While the prior-bias mechanism predicts a multiplicative effect, the value-directed attention mechanism predicts that fixation time and choice will be directly related (as predicted by the additive model). Although we did not see strong evidence for value-directed attention in the binary dataset, such a bias has been shown in explicit information gathering settings [69] and could be at work in other binary choice settings.

Our work most closely relates to two recent lines of work on optimal information sampling for simple choice. First, Hébert and Woodford [70, 71] consider sequential sampling models based on rational inattention. They derive optimal sampling strategies under highly general information-theoretic constraints, and establish several interesting properties of optimal sampling, such as the conditions under which the evidence accumulation will resemble a jump or a diffusion process. In their framework, the decision maker chooses, at each time point, an arbitrary *information structure*, the probability of producing each possible signal under different true states of the world. In contrast, we specify a very small set of information structures, each of which corresponds to sampling a noisy estimate of one item's value (Eq 1). This naturally associates each information structure with fixating on one of the items, allowing us to compare model predictions to human fixation patterns. Whether human attention more closely resembles flexible construction of dynamic information structures, or selection from a small set of fixed information structures is an interesting question for future research.

In a second line of work, concurrent to our own, Jang, Sharma, and Drugowitsch [68] develop a model of optimal information sampling for binary choice with the same Bayesian structure as our model and compare their predictions to human behavior in the same binary choice dataset that we use [9]. There are three important differences between the studies. First, they consider the possibility that samples can also be drawn in parallel for the unattended item, but with higher variance. However, they find that a model in which almost no information is acquired for the unattended item fits the data best, consistent with the assumptions of our model. Second, they use dynamic programming to identify the optimal attention policy almost exactly. This allows them to more accurately characterize truly optimal attention allocation. However, dynamic programming is intractable for more than two items, due to the curse of dimensionality. Thus, they could not consider trinary choice, which is of special interest because only this case makes value-directed attention optimal, and forces the decision-maker to decide which of the unattended items to fixate next, rather than simply when to switch to the other item. Third, they assumed (following previous work) that the prior mean is zero. In contrast, by varying the prior, we show that although a biased prior is needed to account for the attention-choice correlation in binary choice, the data is best explained by a model with only a moderately biased prior mean, about halfway between zero and the empirical mean.

We can also draw insights from the empirical patterns that the model fails to capture. These mismatches suggest that the model, which was designed to be as simple as possible, is missing critical components that should be explored in future work. For example, the underprediction of fixation durations early in the trial could be addressed by more realistic constraints on the fixation process such as inhibition of return, and the overprediction of the proportion of single-fixation trials in the two-item case could be explained with uncertainty aversion. Although not illustrated here, the model's accuracy could be further improved by including bottom-up influences on fixations (e.g., spatial or saliency biases [18, 72]).

While we have focused on attention in simple choice, other studies have explored the role of attention in more complicated multi-attribute choices [5, 73–82]. None of these studies have carried out a full characterization of the optimal sampling process or how it compares to observed fixation patterns, although see [83, 84] for some related results. Extending the methods in this paper to that important case is a priority for future work. Finally, in contrast to many sequential sampling models, our model is not intended as a biologically plausible process

model of how the brain actually makes decisions. Exploring how the brain might approximate the optimal sampling policy presented here, and also how optimal sampling might change under accumulation mechanisms such as decay and inhibition is another priority for future work.

## Methods

The model was implemented in the Julia programming language [85]. The code can be found at https://github.com/fredcallaway/optimal-fixations-simple-choice.

### Attention allocation as a metalevel Markov decision process

To characterize optimal attention allocation in our model, we cast the model as a metalevel Markov decision process (MDP) [52]. Like a standard MDP, a metalevel MDP is defined by a set of states, a set of actions, a transition function giving the probability of moving to each state by executing a given action in a given state, and a reward function giving the immediate utility gained by executing a given action in a given state. In a metalevel MDP, the states, $\mathcal{B}$, correspond to beliefs (mental states), and the actions, $\mathcal{C}$, correspond to computations (cognitive operations). However, formally, it is identical to an MDP, and can be interpreted as such.

In our model, a belief state, $b \in \mathcal{B}$, corresponds to a set of posterior distributions over each item's value. Because the distributions are Gaussian, the belief can be represented by two vectors, $\boldsymbol{\mu}$ and $\boldsymbol{\lambda}$, that specify the mean and precision of each distribution. That is

$$p(u^{(i)} \mid b) = \text{Gaussian}(u^{(i)}; \mu^{(i)}, 1/\lambda^{(i)})$$

To model the switching cost, the belief state must also encode the currently attended item, i.e., the item sampled last (taking a null value, $\oslash$, in the initial belief). Thus, a belief is a tuple $b_t = (\boldsymbol{\mu}_t, \boldsymbol{\lambda}_t, \text{last}_t)$. The dimensionality of the belief space is $2N + 1$ where $N$ is the number of items.

A computation, $c \in \mathcal{C}$, corresponds to sampling an item's value and updating the corresponding estimated value distribution. There are $N$ such computations, one for each item. Additionally, all metalevel MDPs have a special computation, $\perp$ that terminates the computation process (in our case, sampling) and selects an optimal external action given the current belief state (in our case, choosing the item with maximal posterior mean).

The metalevel transition function describes how computations update beliefs. In our model, this corresponds to the sampling and Bayesian belief updating procedure specified in Eq 2, which we reproduce here for the reader's convenience. Note that we additionally make explicit the variable that tracks the previously sampled item. Given the current belief, $b_t = (\boldsymbol{\mu}_t, \boldsymbol{\lambda}_t, \text{last}_t)$, and computation, $c$, the next belief state, $b_{t+1} = (\boldsymbol{\mu}_{t+1}, \boldsymbol{\lambda}_{t+1}, \text{last}_{t+1})$, is sampled from the following generative process:

$$
\begin{aligned}
x_t &\sim \text{Gaussian}(u^{(c)}, \sigma_x^2) \\
\lambda_{t+1}^{(c)} &= \lambda_t^{(c)} + \sigma_x^{-2} \\
\mu_{t+1}^{(c)} &= \frac{\sigma_x^{-2} x_t + \lambda_t^{(c)} \mu_t^{(c)}}{\lambda_{t+1}^{(c)}} \\
\lambda_{t+1}^{(i)} &= \lambda_t^{(i)} \text{ and } \mu_{t+1}^{(i)} = \mu_t^{(i)} \text{ for } i \neq c. \\
\text{last}_{t+1} &= c
\end{aligned}
\tag{6}
$$

Finally, the metalevel reward function incorporates both the cost of computation and the utility of the chosen action. The metalevel reward for sampling is defined

$$R(b_t, c_t) = -\text{cost}(b_t, c_t) = -(\gamma_{\text{sample}} + \mathbf{1}(\text{last}_t \neq \oslash \wedge c_t \neq \text{last}_t)\, \gamma_{\text{switch}}).$$

That is, the cost of sampling includes a fixed cost, $\gamma_{\text{sample}}$, as well as an additional switching cost, $\gamma_{\text{switch}}$, that is paid when sampling from a different item than that sampled on the last time step. We assume that this cost is not paid for the first fixation; however, this assumption has no effect on the optimal policy for reasonable parameter values. The action utility is the true value of the chosen item, i.e., $u^{(i_T^*)}$ where $i_T^* = \underset{i}{\text{argmax}}\ \mu_T^{(i)}$. The metalevel reward for the termination computation, $\perp$, is the expectation of this value. Because we assume accurate priors and Bayesian belief updating, this expectation can be taken with respect to the agent's own beliefs [52], resulting in

$$R(b_t, \perp) = \text{E}[u^{(i_T^*)} \mid b_t] = \max_i \mu_t^{(i)}.$$

## Optimal metalevel policy

The solution to a metalevel MDP takes the form of a Markov policy, $\pi$, that stochastically selects which computation to take next given the current belief state. Formally, $c_t \sim \pi(b_t)$. The optimal metalevel policy, $\pi^*$, is the one that maximizes expected total metalevel reward,

$$\pi^* = \underset{\pi}{\text{argmax}}\ \text{E}\left[\sum_t^T R(b_t, c_t) \mid c_t \sim \pi(b_t)\right].$$

Replacing $R$ with its definition, we see that this requires striking a balance between the expected value of the chosen item and the computational cost of the samples that informed the choice,

$$\pi^* = \underset{\pi}{\text{argmax}}\ \text{E}\left[\max_i \mu_T^{(i)} - \sum_t^{T-1} \text{cost}(b_t, c_t) \mid c_t \sim \pi(b_t)\right].$$

That is, one wishes to acquire accurate beliefs that support selecting a high-value item, while at the same time minimizing the cost of the samples necessary to attain those beliefs. This suggests a strategy for selecting computations optimally. For each item, estimate how much one's decision would improve if one sampled from it (and then continued sampling optimally). Subtract from this number the cost of taking the sample (and also the estimated cost of the future samples). Now identify the item for which this value is maximal. If it is positive, it is optimal to take another sample for this item; otherwise, it is optimal to stop sampling and make a decision.

This basic logic is formalized in rational metareasoning as the *value of computation* (VOC) [51]. Formally, VOC($b$, $c$) is defined as the expected increase in total metalevel reward if one executes a single computation, $c$, and continues optimally rather than making a choice immediately (i.e., executing $\perp$):

$$\text{VOC}(b_t, c) = R(b_t, c) + \text{E}\left[\sum_{t'=t+1}^T R(b_{t'}, c_{t'}) \mid c_{t'} \sim \pi^*(b_{t'})\right] - R(b, \perp).$$

In our model, this can be rewritten

$$\text{VOC}(b_t, c) = -\text{cost}(b_t, c) + \text{E}\left[\max_i \mu_T^{(i)} - \sum_{t'=t+1}^{T-1} \text{cost}(b_{t'}, c_{t'})\,\middle|\; c_{t'} \sim \pi^*(b_{t'})\right] - \max_i \mu_t^{(i)}.$$

That is, the VOC for sampling a given item in some belief state is the expected improvement in the value of the chosen item (rather than making a choice based on the current belief) minus the cost of sampling that item and the expected cost of all future samples.

We can then define the optimal policy as selecting computations with maximal VOC:

$$\pi^*(b) \sim \text{Uniform}(\underset{c}{\text{argmax}}\ \text{VOC}(b, c)).$$

For those familiar with reinforcement learning, this recursive joint definition of $\pi^*$ and VOC is exactly analogous to the joint definition of the optimal policy with the state-action value function, $Q$ [86]. Indeed, $\text{VOC}(b, c) = Q(b, c) - R(b, \perp)$.

Finally, by definition, $\text{VOC}(b, \perp) = 0$ for all $b$. Thus, the optimal policy terminates sampling when no computation has a positive VOC.

## Approximating the optimal policy

For small discrete belief spaces, the optimal metalevel policy can be computed exactly using standard dynamic programming methods such as value iteration or backwards induction [87]. These methods can also be applied to low-dimensional, continuous belief spaces by first discretizing the space on a grid [45], and this approach has recently been used to characterize the optimal fixation policy in binary choice [68]. Unfortunately, these methods are infeasible in the trinary choice case, since the belief space has six continuous dimensions. Instead, we approximate the optimal policy by extending the method proposed in [53]. This method is based on an approximation of the VOC as a linear combination of features,

$$\widehat{\text{VOC}}(b, c; \mathbf{w}) = w_1 \text{VOI}_{\text{myopic}}(b, c) + w_2 \text{VOI}_{\text{item}}(b, c) + w_3 \text{VOI}_{\text{full}}(b) - (\text{cost}(c) + w_4), \quad (7)$$

for all $c \neq \perp$, with $\widehat{\text{VOC}}(b, \perp; \mathbf{w}) = \text{VOC}(b, \perp) = 0$.

We briefly define the features here, and provide full derivations in S1 Appendix. The VOI terms quantify the *value of information* [88] that might be gained by different additional computations. Note that the VOI is different from the VOC because the latter includes the costs of computation as well as its benefits. In general, the VOI is defined as the expected improvement in the utility of the action selected based on additional information rather than the current belief state: $E_{\tilde{b}|b}[R(\tilde{b}, \perp) - R(b, \perp)]$, where $\tilde{b}$ is a hypothetical future belief in which the information has been gained, the distribution of which depends on the current belief.

$\text{VOI}_{\text{myopic}}(b, c)$ denotes the expected improvement in choice utility from drawing one additional sample from item $c$ before making a choice, as opposed to making a choice immediately based on the current belief, $b$. $\text{VOI}_{\text{item}}(b, c)$ denotes the expected improvement from learning the true value of item $c$, and then choosing the best item based on that information. Finally, $\text{VOI}_{\text{full}}(b)$ denotes the improvement from learning the true value of every item and then making an optimal choice based on that complete information.

Together, these three features approximate the expected value of information that could be gained by the (unknown) sequence of future samples. Importantly, this true value of information always lies between the lower bound of $\text{VOI}_{\text{myopic}}$ and the upper bound of $\text{VOI}_{\text{full}}$ (see Fig D in S1 Appendix), implying that the true VOI is a convex combination of these two terms. Note, however, that the weights on this combination are not constant across beliefs, as

assumed in our approximation. Thus, including the $\text{VOI}_{\text{item}}$ term, improves the accuracy of the approximation, by providing an intermediate value between the two extremes. Finally, the last two terms in Eq 7 approximate the cost of computation: cost($c$) is the cost of carrying out computation $c$ and $w_4$ approximates the expected future costs incurred under the optimal policy. Although maximizing $\widehat{\text{VOC}}(b, c; \mathbf{w})$ identifies the policy with the best performance, it is unlikely that humans make attentional decisions using such perfect and noiseless maximization. Thus, we assume that computations are chosen using a Boltzmann (softmax) distribution [55] given by

$$\pi(c \mid b; \mathbf{w}, \beta) \propto \exp\left\{\beta\widehat{\text{VOC}}(b, c; \mathbf{w})\right\},$$

where the inverse temperature, $\beta$, is a free parameter that controls the degree of noise. Note that computation selection is fully random when $\beta = 0$ and becomes deterministic as $\beta \to \infty$.

To identify the weights used in the approximation, we first assume that $w_i \geq 0$ and $w_1 + w_2 + w_3 = 1$, since $w_{1:3}$ features form a convex combination and $w_4$ captures the non-negative future cost. Previous work [53] used Bayesian optimization to identify the weights within this space that maximize total expected metalevel reward. However, we found that often a large area of weight space resulted in extremely similar performance, despite inducing behaviorally distinct policies. Practically, this makes identifying a unique optimal policy challenging, and theoretically we would not expect all participants to follow a single unique policy when there is a wide plateau of high-performing policies. To address this, we instead identify a set of near-optimal policies and assume that human behavior will conform to the aggregate behavior of this set.

To identify this set of near-optimal policies, we apply a method based on Upper Confidence Bound (UCB) bandit algorithms [89]. We begin by sampling 8000 weight vectors to roughly uniformly tile the space of possible weights. Concretely, we divide a three-dimensional hypercube into $800 = 20^3$ equal-size boxes and sample a point uniformly from each box. The first two dimensions are bounded in (0, 1) and are used to produce $w_{1:3}$ using the following trick: Let $x_1$ and $x_2$ be the lower and higher of the two sampled values. We then define $w_{1:3} = [x_1, x_2 - x_1, 1 - x_2]$. If $x_1$ and $x_2$ are uniformly sampled from (0, 1), and indeed they are, then this produces $w_{1:3}$ uniformly sampled from the 3-simplex. The third dimension produces the future cost weight; we set $w_4 = x_3 \cdot$ maxcost where maxcost is the lowest cost for which no computation has positive $\widehat{\text{VOC}}$ in the initial belief state. We then simulate 100 decision trials for each of the resulting policies, providing a baseline level of performance. Using these simulations, we compute an upper confidence bound of each policy's performance equal to $\hat{\mu}_i + 3\hat{\sigma}_i$, where $\hat{\mu}_i$ and $\hat{\sigma}_i$ are the empirical mean and standard deviations of the metalevel returns sampled for policy $i$. A standard UCB algorithm would then simulate from the policy maximizing this value. However, because we are interested in identifying a set of policies, we instead select the top 80 (i.e. 1% of) policies and simulate 10 additional trials for each, updating $\hat{\mu}_i$ and $\hat{\sigma}_i$ for each one. We iterate this step 5000 times. Finally, we select the 80 policies with the highest expected performance as our characterization of optimal behavior in the metalevel MDP. To eliminate the possibility of fitting noise in the optimization procedure, we use one set of policies to compute the likelihood on the training data and re-optimize a new set of policies to generate plots and compute the likelihood of the test data. Note that we use the box sampling method described in the previous paragraph rather than a deterministic low discrepancy sampling strategy [90] so that the set of policies considered are not exactly the same in the fitting and evaluation stages.

How good is the approximation method? Previous work found that this approach generates near-optimal policies on a related problem, with Bernoulli-distributed samples and no

switching costs [53]. Note that in the case of Bernoulli samples, the belief space is discrete and thus the optimal policy can be computed exactly if an upper bound is placed on the number of computations that can be performed before making a decision. Although introducing switching costs makes the metareasoning problem more challenging to solve, in the Bernoulli case we have found that they only induce a modest reduction in the performance of the approximation method relative to the full optimal policy, achieving 92% of optimal reward in the worst case (see S1 Appendix for details). This suggests that this method is likely to provide a reasonable approximation to the optimal policy in the model with Gaussian samples used here, but a full verification of this fact is beyond the scope of the current study.

## Implementation of the prior

In the main text, we specified the prior as a property of the initial belief state. However, for technical reasons (in particular, to reuse the same set of optimized policies for multiple values of $\alpha$), it is preferable to perform policy optimization and simulation in a standardized space, in which the initial belief state has $\boldsymbol{\mu}_0 = \mathbf{0}$ and $\boldsymbol{\lambda}_0 = \mathbf{1}$. We then capture the prior over the ratings of items in the experiment by transforming the ratings into this standardized space such that the transformed values are in units defined by the prior. Concretely, given an item rating $r^{(i)}$, we set the true value to

$$u^{(i)} = \frac{r^{(i)} - \bar{\mu}}{\bar{\sigma}}, \tag{8}$$

where $\bar{\mu}$ and $\bar{\sigma}$ denote the prior mean and standard deviation. Modulo the resultant change in units (all parameter values are divided by $\bar{\sigma}$), this produces the exact same behavior as the naïve implementation, in which the initial belief itself varies.

There is one non-trivial consequence of using this approach when jointly fitting multiple datasets: The jointly fit parameters are estimated in the standardized space, rather than the space defined by the raw rating scale. As a result, if we transform the parameters back into the raw rating space, the parameters will be slightly different for the two datasets (even though they are identical in the transformed space). This was done intentionally because we expect that the parameters will be consistent in the context-independent units (i.e., standard deviations of an internal utility scale). However, this decision turns out to have negligible impact in our case because the empirical rating distributions are very similar. Specifically, the empirical rating distributions are (mean ± std) 3.492 ± 2.631 for the binary dataset and 4.295 ± 2.524 for the trinary dataset. Due to the difference in standard deviations, all parameters (except $\alpha$, which is not affected) are 2.631/2.524 = 1.042 times larger in the raw rating space for the binary dataset compared to the trinary dataset. The difference in empirical means affects $\bar{\mu}$, which is 3.492/4.295 = 0.813 times as large in the binary compared to trinary dataset. However, given our interpretation of $\alpha$ as a degree of updating towards the empirical mean, this difference is as intended.

## Model simulation procedure

Given a metalevel MDP and policy, $\pi$, simulating a choice trial amounts to running a single episode of the policy on the metalevel MDP. To run an episode, we first initialize the belief state, $b_0 = (\boldsymbol{\mu}_0 = \mathbf{0}, \boldsymbol{\lambda}_0 = \mathbf{1}, \text{last}_0 = \oslash)$. Note that $\text{last}_0 = \oslash$ indicates that no item is fixated at the onset of a trial.

The agent then selects an initial computation $c_0 \sim \pi(b_0)$ and the belief is updated according to the transition dynamics (Eq 6). Note that $\pi(c|b_0)$ assigns equal sampling probability to all of the items, since the subject starts with symmetrical beliefs. This process repeats until some

time step, $T$, when the agent selects the termination action, $\perp$. The predicted choice is the item with maximal posterior value, $i_T^* = \underset{i}{\mathrm{argmax}}\ \mu_T^{(i)}$. In the event of a tie, the choice is sampled uniformly from the set of items with maximal expected value in the final belief state; in practice, this never happens with well-fitting parameter values.

To translate the sequence of computations into a fixation sequence, we assume that each sample takes 100 ms and concatenate multiple contiguous samples from the same item into one fixation. The temporal duration of a sample is arbitrary; a lower value would result in finer temporal predictions, but longer runtime when simulating the model. In this way, it is very similar to the $dt$ parameter used in simulating diffusion decision models. Importantly the qualitative predictions of the model are insensitive to this parameter because $\sigma_x$ and $\gamma_{\text{sample}}$ can be adjusted to result in the same amount of information and cost per ms.

We simulate the model for two different purposes: (1) identifying the optimal policy and (2) comparing model predictions to human behavior. In the former case, we randomly sample the true utilities on each "trial" i.i.d. from Gaussian(0, 1). This corresponds to the assumption that the fixation policy is optimized for an environment in which the DM's prior is accurate. When simulating a specific trial for comparison to human behavior, the true value of each item is instead determined by the liking ratings for the items presented on that trial, as specified in Eq 8.

## Model parameter estimation

The model has five free parameters: the standard deviation of the sampling distribution, $\sigma_x$, the cost per sample, $\gamma_{\text{sample}}$, the cost of switching attention, $\gamma_{\text{switch}}$, the degree of prior updating, $\alpha$, and the inverse temperature of the Boltzmann policy, $\beta$. We estimate a single set of parameters at the group level using approximate maximum likelihood estimation in the combined two- and three-item datasets, using only the even trials.

To briefly summarize the estimation procedure: given a candidate set of parameter values, we construct the corresponding metalevel MDP and identify a set of 80 near-optimal policies for that MDP. We then approximate the likelihood of the human fixation and choice data using simulations from the optimized policies. Finally, we perform this full procedure for 70,000 quasi-randomly sampled parameter configurations and report the top thirty configurations (those with the highest likelihood) to give a rough sense of the uncertainty in the model predictions. A parameter recovery exercise (reported in S1 Appendix) suggests that this method, though approximate, is sufficient to identify the parameters of the model with fairly high accuracy. Below, we explain in detail how we estimate and then maximize the approximate likelihood.

The primary challenge in fitting the model is in estimating the likelihood function. In principle, we could seek to maximize the joint likelihood of the observed fixation sequences and choices. However, like most sequential sampling models, our model does not have an analytic likelihood function. Additionally, the high dimensionality of the fixation data makes standard methods for approximating the likelihood [91, 92] infeasible. Thus, taking inspiration from Approximate Bayesian Computation methods [93, 94], we approximate the likelihood by collapsing the high dimensional fixation data into four summary statistics: the identity of the chosen item, the number of fixations, the total fixation time, and the proportion of fixation time on each item. As described below, we estimate the joint likelihood of these summary statistics as a smoothed histogram of the statistics in simulated trials, and then approximate the likelihood of a trial by the likelihood of its summary statistics. We emphasize, however, that we do not use this approximate likelihood to evaluate the performance of the model. Instead, we intend it to be a maximally principled (and minimally researcher-specified) approach to

choosing model parameters, given that computing a true likelihood is computationally infeasible.

Given a set of near-optimal policies, we estimate the likelihood of the summary statistics for each trial using a smoothed histogram of the summary statistics in simulated trials. Critically, this likelihood is conditional on the ratings for the item in that trial. However, it depends only on the (unordered) set of these ratings; thus, we estimate the conditional likelihood once for each such set. Given a set of ratings, we simulate the model 625 times for each of the 80 policies, using the resulting 50,000 simulations to construct a histogram of the trial summary statistics. The continuous statistics (total and proportion fixation times) are binned into quintiles (i.e., five bins containing equal amounts of the data) defined by the distribution in the experimental data. For the fixation proportions, the quintiles are defined on the rating rank of the item rather than the spatial location because we expect the distributions to depend on relative rating in the three-item case. Values outside the experimental range are placed into the corresponding tail bin. Similarly, trials with five or more fixations are all grouped into one bin (including e.g., six and seven fixations) and cases in which the model predicts zero fixations are grouped into the one-fixation bin. This latter case corresponds to choosing an item immediately without ever sampling, and occurs rarely in well-fitting instantiations of the model, but happens frequently when $\gamma_{\text{sample}}$ is set too high. For each simulation, we compute the binned summary statistics, identify the corresponding cell in the histogram, and increase its count by one. Finally, we normalize this histogram, resulting in a likelihood over the summary statistics. To compute the likelihood of a trial, $\mathcal{L}(d \mid \theta)$, we compute the binned summary statistics for the trial and look up the corresponding value in the normalized histogram for that trial's rating set.

To account for trials that are not well explained by our model, we use add-$n$ smoothing, where $n$ was chosen independently for each $\theta$ to maximize the likelihood. This is equivalent to assuming a mixture between the empirical distribution and a uniform distribution with mixing weight $\epsilon$. Thus, the full approximate likelihood is

$$\mathcal{L}(D \mid \theta) = \max_{\epsilon \in [0, 0.5]} \prod_{d \in D} \left( \epsilon \frac{1}{C} + (1 - \epsilon) \, \mathcal{L}(d \mid \theta) \right),$$

where $C = N \cdot 5^{N+1}$ is the total number of cells in the histogram. Importantly, this error model is only used to approximate the likelihood; it is not used for generating the model predictions in the figures—indeed, it could not be used in this way because the error model is defined over the summary statistics, and cannot generate full sequences of fixations. Thus, the $\epsilon$ parameter should be interpreted in roughly the same way as the bandwidth parameter of a kernel density estimate [91], rather than as an additional free parameter of the model.

We then use this approximate likelihood function to identify a maximum likelihood estimate, $\hat{\theta} = \arg \max \mathcal{L}(D \mid \theta)$. Based on manual inspection, we identified the promising region of parameter space to be $\sigma_x \in (1, 5)$, $\gamma_{\text{sample}} \in (0.001, 0.01)$, $\gamma_{\text{switch}} \in (0.003, 0.03)$, and $\beta \in (100, 500)$. We then ran an additional quasi-random search of 10,000 points within this space using Sobol low-discrepancy sequences [90]. This approach has been shown to be more effective than both grid search and random search, while still allowing for massive parallelization [95].

Note that the optimal policy does not depend on $\alpha$ because the DM believes her prior to be unbiased (by definition) and makes her fixation decisions accordingly. The alternative, optimizing the policy conditional on $\alpha$, would imply that the DM is internally inconsistent, accounting for the bias in her fixations but not in the prior itself. Thus, we optimize $\alpha$ separately from the other parameters. Specifically, we consider 10,000 possible instantiations of all the other parameters, find optimal policies once for each instantiation, and evaluate the

likelihood for seven values of $\alpha$; these seven values included the special cases of 0 and 1 as well as five additional evenly-spaced values with a random offset (roughly capturing the low-discrepancy property of the Sobol sequence).

We found that the stochasticity in the policy optimization and likelihood estimation coupled with weak identifiability for some parameters resulted in slightly different results when re-running the full procedure; thus, to give a rough sense of the uncertainty in the estimate, we identify the top thirty parameters, giving us both mean and standard deviation for each parameter and the total likelihood.

## Supporting information

**S1 Appendix. Supplementary methods and results.** Includes descriptions of the tasks for the datasets we model, individual fitting methods and summary of results, parameter recovery results, aDDM implementation and validation, derivations for the value of information features, and a validation of the policy approximation method.
(PDF)

**S2 Appendix. Individual fitting results.** Includes versions of all plots in the main text with separate panels for each participant (including model predictions with parameters fit to each participant).
(PDF)

## Acknowledgments

We thank Ian Krajbich for his help in simulating the aDDM and Bas van Opheusden for suggesting the method for efficiently computing $\text{VOI}_{\text{full}}$.

## Author Contributions

**Conceptualization:** Frederick Callaway, Antonio Rangel, Thomas L. Griffiths.

**Data curation:** Antonio Rangel.

**Formal analysis:** Frederick Callaway.

**Funding acquisition:** Thomas L. Griffiths.

**Software:** Frederick Callaway.

**Supervision:** Antonio Rangel, Thomas L. Griffiths.

**Visualization:** Frederick Callaway.

**Writing – original draft:** Frederick Callaway, Antonio Rangel.

**Writing – review & editing:** Frederick Callaway, Antonio Rangel, Thomas L. Griffiths.

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
