## [Decision Letter · Decision Letter 0]

15 Oct 2020

Dear Callaway,

Thank you very much for submitting your manuscript "Fixation patterns in simple choice reflect optimal information sampling" for consideration at PLOS Computational Biology.

As with all papers reviewed by the journal, your manuscript was reviewed by members of the editorial board and by several independent reviewers. In light of the reviews (below this email), we would like to invite the resubmission of a significantly-revised version that takes into account the reviewers' comments (major revisions).

All Reviewers' points need to be addressed very carefully. We will send the revisions back to the original reviewers.

I would like to stress that both Reviewer 2 and Reviewer 3 raise important issues concerning the model assumption (DDM vs. aDDM and sequential vs. parallel sampling) that will require additional and extensive modelling work.

We cannot make any decision about publication until we have seen the revised manuscript and your response to the reviewers' comments. Your revised manuscript is also likely to be sent to reviewers for further evaluation.

Sincerely,

Stefano Palminteri

Associate Editor

PLOS Computational Biology

Samuel Gershman

Deputy Editor

PLOS Computational Biology

Reviewer's Responses to Questions

**Comments to the Authors:**

Reviewer #1: In the manuscript entitled “Fixation patterns in simple choice reflect optimal information sampling”, Callaway and colleagues propose a model for optimal information sampling in value-based decision making. The model is fitted to binary and trinary choices data, and data fit and misfits are discussed.

Overall, I found this paper quite interesting. It raises an important issue about the optimality of fixations in value-based decision making that has not been explored so far. However, I am concerned about the theoretical assumptions of such a model.

Main points

1. My first concern is about the concept of information sampling in value-based decision making and its dependency on visual inputs. While visual inputs constitute the accumulated evidence in perceptual decision making (as described in the sequential sampling framework), the evidence accumulated in value-based decisions has less to do with visual aspects of the stimuli (as they are typically easy to identify based on their appearances) and more with values retrieved from memory. In fact, value based decisions can also be carried out without any visual input: One could ask participants to press left if they prefer a Kit-kat and right if they prefer a Bounty, without presenting Kit-kats and Bounties on the left and right sides of the screen. Participants could also be asked to choose between a right and a left option and to fixate a cross in the middle of the screen. Therefore, fixations should not be seen as instrumental or necessary for value based-decisions. Because of this, the authors’ position, i.e., that “fixations are deployed to sample information optimally in order to make the best possible choice” [line 39] is to me inappropriate in value-based decisions. The authors say that such an account comes from perceptual decision making, where fixations are instrumental for the accumulation of evidence (the more attention to a specific area of the visual field, the better discriminating evidence I can gather). This is however not the case for value-based decisions, as it is possible to gather information for or against one option without looking at it (because the evidence is stored in memory).

2. The other, related, theoretical concern has to do with the use of “selective/sequential”, as opposed to “parallel” information sampling. The authors say that, in the DDM, “the same amount of information about all options” is received “in parallel” [line 60], and the DDM constitutes the optimal algorithm for that kind of situation. However, in the aDDM, “decisions are based on sequentially accumulated value samples” [line 31] and that “the optimal algorithm when the decision maker can sample information selectively is unknown” [line 62]. In their model, participants get samples only from the item they are gazing at, and can only get samples from the other item(s) by switching the gaze towards them, which is costly. I have 3 points related this:

(a) can the author actually provide evidence that participants are indeed gathering information selectively/sequentially? This would mean to find evidence that people only gather information about an option at a time. This seems to be a crucial point: If there is actually no evidence for this, then the DDM is actually still the optimal algorithm (well, at least for 2-options tasks).

(b) The other point is that there should be some clarification about mutual inhibition. In both the DDM and aDDM, evidence is accumulated as a single sum, and there is perfect inhibition between the 2 options. This means that the higher the evidence in favour of one option at a time point t (including the noise), the more discounted the evidence for the other option will be. As far as I understand (please correct me if I am wrong), in their account, it seems like there is no inhibition at all. So I think what the authors are proposing here is not only a model sequential information sampling, but also information sampling in which the evidence accumulated for one option is independent from the one accumulated from the other. I think it should be an open question whether this is optimal in the case of sequential information sampling as well. In case I misunderstood this point, I would still like the question of inhibition to be clarified in the paper, explaining how that plays a crucial role in optimality.

(c) I think it is incorrect to say that in the aDDM samples are accumulated sequentially. In the aDDM, the drift-rate changes within a trial to account for attentional fluctuations. However, the drift-rate is the same as in the DDM: it assumes that evidence is accumulated in a single sum and with perfect inhibition, so never only for one option alone.

3. In a previous paper (Smith & Krajbich, Psychol Sci, 2018) the aDDM was compared with a model in which "gaze merely adds evidence, providing a fixed advantage for the attended option”, also called by the authors an additive model (Krajbich, Current Opinion in Psychology, 2018). The authors found that only the aDDM could account for overall-value effects. Can the author explain what their optimal model say about such effects? Also can the authors discuss the similarities between theirs and this additive model accounts?

4. I think it would benefit to report the aDDM fits (quantitative and qualitative) in this paper as well, even though they might be found in the original papers. These could work well as a benchmark model and make very clear where the authors’ proposed account and the aDDM converge/diverge.

5. In my opinion, the authors should provide a better definition of optimality in value-based decision making. Can their view be better grounded in existing literature perhaps?

6. I found the Methods a bit brief and at the same time quite hard to read. I suggest to make them a bit more accessible. I have a few points here:

(a) The model is fit at a group level, across two tasks, and only group-level patterns are shown for the model fits. Can the authors justify this choice and also show how individual fits actually look like? As said before, perhaps comparing their model to the aDDM?

(b) Can the authors provide evidence that the model is identifiable and that the parameters (in this case at the group level) are unbiased? Is the use of the chosen trial summary statistics for fitting sufficient and are they unbiased?

Minor points

- Figure 3B: Better use quantiles and not kernel density estimators to compare RT/fixation time distributions.

- Response times (versus reaction), as we are in the value-based and not perceptual domain.

- I believe the (more) correct name is diffusion decision model (and not drift diffusion model), see for example: Ratcliff, R., Smith, P. L., Brown, S. D., & McKoon, G. (2016). Diffusion decision model: Current issues and history. Trends in cognitive sciences, 20(4), 260-281.

- Repetition of “process”, line 29.

- "choice" is missing, line 81.

- Something funny on line 512.

- Quantiles spelled wrong, line 533.

Reviewer #2: ## Summary

The manuscript presents an "optimal" joint model for choices, fixations, and fixation times in binary and ternary value-based decision making. The model is an optimal model in the sense that it assumes that an agent attempts to maximize their payoff while choosing from which of the options to sample (i.e., fixate on) which increases the information of the value of the option and when to stop and pick the option with the highest value. The model has only four free parameter (or five parameters, see below), but is able to predict an impressive number of empirical patterns and only fails to explain a few, especially for the binary choices. Importantly, the model performance is evaluated using 50% of hold-out data that was not used to fit the model.

## Evaluation

There is a lot to like about this paper: It starts with an interesting perspective of the task of the participant, considers several data types (i.e., eye movements and choices), implements the idea in a formally and technically sophisticated manner, uses different data for model estimation and model testing, and the model provides an overall pretty good performance. Thus, on the positive side, the paper provides an impressive computational effort to value-based decision making. However, there are also a few negative issues, listed below, that somewhat limit the impact of the paper. The big lingering question is what we really learn from the model about the task? Thus, my main suggestions are about the framing of the results. Nevertheless, I believe these issues can be fixed in a revision by providing a more balanced view of the contribution of the model. In addition, there are a few situation which I believe some parts could be clarified a bit.

Main issues

1. This might be more of a reviewer than an author issue, but I have some problems with the usage of the term "optimal" in this manuscript. To me, the term "optimal model" implies that the model captures the relevant constraints of the task and implements the one strategy (or "metalevel" strategy in the present case) that, if followed, maximises the payoff. However, in the current case we now have the surprising situation that the data is largely consistent with the optimal model for the three item case, but shows some more pronounced misfit for the two item case. What does this mean? One possibility is that participants do not follow the optimal strategy in the two item case, but they do follow it in the three item case. I find this possibility hard to believe. What would be the reason that when going from two to three items the behaviour of the participants changes so substantially? The other possibility seems to be, and this is also the possibility explicitly mentioned by the authors in the discussion (ll. 345), that the optimal model is not really the optimal model and other optimal models could capture the pattern for both two and three items. But if this is the case, in what sense is then the original model actually optimal? In other words, what is the benefit of avoiding proposing a "specific algorithmic model" if the optimal model is treated in exactly the same manner as such an algorithmic model, its assumption are subject to change if they do not fit the data (for a discussion see e.g., Tauber et al., 2017, Psychological Review).

2. A somewhat related point (i.e., also discussed in Tauber et al., 2017) is the question of individual behaviour versus aggregate behaviour. The approach of the authors of fitting a single parameter set across two experiments and showing that the model simultaneously predicts both data sets is pretty impressive. However, the downside of this is that all evaluations of the model also happens on the aggregate level and are prone to the ecological fallacy. There is no indication whether any of the data pattern against which the model is compared is a pattern that holds across individuals, for a group of individuals, or is potentially an aggregation artefact. I do not think this disqualifies the current model but this issue clearly restricts its explanatory power. I feel like that the two approaches to handle it are either providing some indication for each data pattern how strongly it holds across participants (e.g., graphically if possible) or acknowledge this limitation explicitly.

3. One of the key take-home messages of the paper that the authors highlight themselves is that "the results show that the fixation process during simple choice is influenced dynamically by the value estimates computed during the decision process" (quote from the abstract). It is clear that this is an important insight going forward. However, I wonder how much of this take away is a consequence of the model. In contrast, the discussion highlights that this is mainly a behavioral pattern (see ll. 339 to 341) that needs to be explained. The current model does so, but it is unclear if it really is the only one that does so. I think it would be helpful to be clearer in how much of this insight is data driven and how much is model driven.

4. The main text states (p. 7, 164): "We use maximum likelihood to fit these parameters to choice and fixation data." I think this sentence is an overly generous summary of the actual fitting process. It is clear that this model and the fitting process have a lot of moving parts and fitting this model requires a lot of choices. One obvious reason for this is that the complexity of the model is such that only a stochastic approach for fitting the model is computationally feasible. However, instead of using a more established approach for fitting such models (e.g., ABC, Palestro et al., 2019; likelihood-free methods, e.g., van Opheusden et al., 2020) the authors develop their own method for fitting such data which involves steps of data transformation and binning and a grid-search to find plausible parameter regions. The authors provide some assurances that this method is valid, especially for a simplified case, but we mostly have to take the authors word that they implemented everything correctly and none of the somewhat arbitrary choices plays a too important role. Of course, the biggest argument for the authors case is the overall good performance of the model on the 50% hold-out data. To sum this up, I think one could argue if the maximum likelihood for this model and data actually exists (e.g., what is the conditional distribution of the fixation times?). To me it looks as if the authors have found a set of representative parameter estimates that produce the main qualitative patterns in the data. Maybe one could make a case that the procedure "approximates" the likelihood or "pseudo-likelihood" but even this seems a bit of a stretch.

5. I had problems understanding the "Uncertainty-directed attention" section of the paper. In particular, I did not understand what the actual DV is here. For example, why is a negative "relative cumulative fixation time at the beginning of a new fixation, starting with the second fixation" an indication for "a tendency to fixate on items that have been fixated relatively less in the trial"? Again, this might be more of a problem that I am missing the relevant background, but this was the only section of the paper in which I did not understand the measures and how they were derived from the data. Maybe this can be clarified.

6. The method section contains a section on the "Optimal policy for model with random fixations". However, this policy does not any more seem to be part of the paper. Maybe it was part of a previous version?

7. Again, this probably shows my lack of relevant background knowledge, but it may be helpful to add an explanation of what the difference between a metalevel MDP and a regular MDP for the actual fitting is. Does it make a difference if I simulate from a regular MDP or a metalevel MDP? If so, how?

8. It is unclear if epsilon, the mixture weight between the approximate likelihood and the uniform error process, is also a free parameter. From the description it it seems like it, then it would be more fair to say the model as five free parameters. Also, how big is the estimate of epsilon?

9. The method section states "thus, to give a rough sense of the uncertainty in the estimate, we identify the top thirty parameters, giving us both mean and standard error estimates for each parameter and the total likelihood." The main text gives the parameter estimates as "mean ± std". Please clarify: Is the "std" in the main text the standard error of the parameter estimates (i.e., SD of the 30 top estimates divided by sqrt(30)) or simply the SD of the 30 top estimates? If the former I am not sure I understand how it makes sense to divide by sqrt(30) here. The 30 seems to ba an arbitrary cut-off point (to denote something like "close enough") but this would then not justify to use this as the sample size in the standard error calculation.

10. The method section states "Based on manual inspection, we identified the promising region of parameter space to be ... A grid search confirmed that this area was likely to contain the global optimum (see Fig. S4)." However, with the exception of one parameter, y_switch, figure S4 only shows the range given in this part of the text. Thus, it does not permit the inference that the chosen region is the promising region as essentially no information about values outside the region are available.

Minor issues:

- l. 81: "during simple is influenced". Seems like "decision making" is missing here after "simple".

- Figure 2, note: "This probability correlates strongly with the value of sampling the item". I wonder why it says "correlates strongly". What is missing from a perfect correlation (only the switching costs)?

- l. 512: "liking ratings provided by the participants." seems to be wrong here.

- l. 547: "between" appears twice.

## References:

Palestro, J. J., Sederberg, P. B., Osth, A. F., Zandt, T. V., & Turner, B. M. (2019). Likelihood-Free Methods for Cognitive Science (Softcover reprint of the original 1st ed. 2018 Edition). Springer.

van Opheusden, B., Acerbi, L., & Ma, W. J. (2020). Unbiased and Efficient Log-Likelihood Estimation with Inverse Binomial Sampling. ArXiv:2001.03985 [Cs, q-Bio, Stat]. http://arxiv.org/abs/2001.03985

Reviewer #3: It has been a pleasure to read the paper from Callaway and colleagues on fixation patterns in value-based choice and optimal information sampling. I think this is a very timely paper. In the last decade many studies have investigated the role of attention in value-based choice using models like the aDDM, LCA or the model from Cavanagh et al. However, a shortcoming/oddity of these models is that attention is treated as an exogenous variable and therefore does not provide a satisfactory explanation of where the attention pattern originated. This work attempts to fill in this gap by proposing a generative model of attention based on optimal information sampling, studying its effect on value based choice in binary and trinary decisions. The manuscript is well written and the authors present the results of their data fitting to previous studies in a straightforward way. I particularly appreciated the fact that the authors point out to the reader the cases where their model undershoots, overshoots, or completely fails to predict some of the data. Below are a few suggestions that I hope the authors will find useful in their revisions:

1) It looks like that their model works at its best for 3 options setting (but note that also in this case it is often off mark - e.g. fig 6.B). In fact, while the model captures some of the basic behavioural tendencies for the 2 options task, it completely fails to predict some of the key effects for which the aDDM was developed in the first place (i.e. value-directed attention and choice bias). I appreciate the approach of developing a model with minimal assumptions. But these failures in the 2-choice setting are not just small wrinkles for a model of attention as optimal information sampling. While the authors acknowledge this shortcoming, they don’t provide a compelling explanation of what causes this and how this can be fixed. They mention in the discussion another approach (used in another recent study) that uses a negatively biased prior. But they say in the supplemental that they tried a similar approach “was not sufficient to even qualitatively capture the bias in the binary choice”. However, eyeballing the figure S1B compared with 7B it looks that some of the shape in the data is captured by using biased priors. Is using biased prior the only way to fix this problem or the authors have other intuitions that can help for the binary case? In any case this issue needs to be investigated in more details. Unfortunately the behavioural effect of choice bias and value-directed attention are robust findings at the core of many previous papers using aDDM. I therefore think that this shortcoming cannot be easily swept under the carpet. Models with minimal assumption are fine, but not if they fail to capture (even qualitatively) such strong effects. My fears are double here: 1) such mismatch between data and model might be a clear indication that the authors are missing some key ingredients in their assumptions 2) that the paper will not be not so impactful (or even worst dismissed) by people in the field of DM since most studies use binary choice. Since a lot of good work has gone into this, that would be a real shame. Therefore, I suggest putting more work into understanding what causes these mismatch and how think how it can be ameliorated (even if this comes at a cost of increasing model complexity).

2) For the 3 options experiment most analyses are based on a specific definition of DV X - mean of other 2 options. While this approach has been used in some previous papers, I find it a bit arbitrary. Who decides that participants are evaluating that item against the mean of the other 2 is the right operation? why not the median or a weighted average? or is just considering 2nd best ignoring completely the worst item?. I suggest a more agnostic approach, using multinomial logistic regression model in which the value of each option is inputted independently and not requiring a priori specification of DV.

3) Sum value (i.e. the value of both option) has been used in a number of studies including some recent work from Krajbich group. This has enabled the authors to distinguish between different competing dynamic models (in particular arbitrating between additive vs multiplicative effect of attention). I am wondering if is worth investigating what their optimal model predict for sum value effect on attention. That might give some interesting new insights.

4)The work by Hébert and Woodford 2019 seems relevant here but is not currently discussed.

**Have all data underlying the figures and results presented in the manuscript been provided?**

Reviewer #1: Yes

Reviewer #2: Yes

Reviewer #3: Yes

PLOS authors have the option to publish the peer review history of their article (what does this mean?). If published, this will include your full peer review and any attached files.

Reviewer #1: No

Reviewer #2: No

Reviewer #3: No
---

## [Decision Letter · Decision Letter 1]

18 Feb 2021

Dear Dr Callaway,

Thank you very much for submitting your manuscript "Fixation patterns in simple choice reflect optimal information sampling" for consideration at PLOS Computational Biology. As with all papers reviewed by the journal, your manuscript was reviewed by members of the editorial board and by several independent reviewers. The reviewers appreciated the attention to an important topic. Based on the reviews, we are likely to accept this manuscript for publication, providing that you modify the manuscript according to the review recommendations.

As you will see the Reviewers were overall satisfied by the revisions. There are few remaining issues/suggestions from R1 and R2 that need to be addressed before we can proceed accepting the paper.

Sincerely,

Stefano Palminteri

Associate Editor

PLOS Computational Biology

Samuel Gershman

Deputy Editor

PLOS Computational Biology

[LINK]

Dear Dr Callaway,

As you will see the Reviewers were overall satisfied by the revisions. There are few remaining issues/suggestions from R1 and R2 that need to be addressed before we can proceed accepting the paper.

t

Reviewer's Responses to Questions

**Comments to the Authors:**

Reviewer #1: I thank the authors for carefully replying to my previous points. I believe the manuscript has truly improved.

1. I would like to clarify the issue with the term "sequential". What I meant is that both the DDM and aDDM are sequential sampling models in which evidence is sampled (sequentially) in a relative fashion, so it is always evidence for A relative to B that it sampled (as in the Sequential probability ratio test). This is what distinguishes sequential models in which evidence is accumulated "in a single sum" (such as DDM, aDDM, DFT, OU models) from sequential sampling models in which evidence is accumulated in separate accumulators (such as race models, LCA, etc). Therefore, I would like to stress that when the authors were using the term "sequential" to refer to the fact that evidence was accumulated sequentially for different options, this does not correspond to neither what the DDM or the aDDM "do".

Therefore, I suggest to change the term "sequential" with perhaps "time-varying accumulation rates" or something similar. I think it should be clear that the authors are not suggesting that when looking at A you are not accumulating evidence for B (again, the accumulated evidence is relative for A vs. B, not absolute for A in both the DDM and aDDM). However, the crucial addition of the aDDM vs. the DDM is that it allows the accumulation rate to vary within the trial based on attentional shifts.

2. The other clarification is about inhibition. I am not sure what the authors mean by "In the traditional two-alternative choice DDM (and the original aDDM) the role of inhibition depends on the exact neural network used to implement or approximate the algorithm". In Bogacz 2006 (The Physics of Optimal Decision Making: A Formal Analysis of Models of Performance in Two-Alternative Forced-Choice Tasks) it is shown how the DDM relates to other sequential sampling models only when inhibition is high or there is perfect inhibition. Moreover, since the DDM is related to the SPRT, I am not sure I get why the author say that "Adding inhibition would necessarily violate the rational norm of Bayesian inference, and thus does not seem appropriate for our model given our emphasis on optimality". For what I understand, mutual inhibition is necessary for optimality in sequential sampling models. Perhaps the authors can elaborate/better explain their point.

Reviewer #2: I feel the revision addresses my concerns with the previous version adequately. Overall it is a really interesting paper. I found only two small minor issues listed below.

- l. 366: "In the two item case, the model simulations show a smaller but *also* positive effect." The "but" alone seems to indicate the positive effect contrasts with the trinary case which however is also positive.

- ll. 367 to 369: "This is counterintuitive since the model predicts that in the two-item case fixation locations are insensitive *to* the sign of the relative value estimates (Fig 2A). However, the patter*n* likely arises due to the tendency to fixate last on the chosen item (see Fig 7A below)."

Reviewer #3: The revisions that the authors have conducted address my previous comments. I am therefore happy to recommend this paper for publication in Plos Comp Biology

**Have all data underlying the figures and results presented in the manuscript been provided?**

Reviewer #1: Yes

Reviewer #2: Yes

Reviewer #3: None

PLOS authors have the option to publish the peer review history of their article (what does this mean?). If published, this will include your full peer review and any attached files.

Reviewer #1: No

Reviewer #2: No

Reviewer #3: No

Figure Files:

Data Requirements:

Reproducibility:

References:

---

## [Editor Report · Decision Letter 2]

10 Mar 2021

Dear Mr Callaway

We are pleased to inform you that your manuscript 'Fixation patterns in simple choice reflect optimal information sampling' has been provisionally accepted for publication in PLOS Computational Biology.

Best regards,

Stefano Palminteri

Associate Editor

PLOS Computational Biology

Samuel Gershman

Deputy Editor

PLOS Computational Biology

---

## [Editor Report · Acceptance letter]

23 Mar 2021

PCOMPBIOL-D-20-01568R2 

Fixation patterns in simple choice reflect optimal information sampling

Dear Dr Callaway,

I am pleased to inform you that your manuscript has been formally accepted for publication in PLOS Computational Biology. Your manuscript is now with our production department and you will be notified of the publication date in due course.

With kind regards,

Andrea Szabo
